# Group Equivariant Subsampling

**Jin Xu**[1*]    **Hyunjik Kim**[2]    **Tom Rainforth**[1]    **Yee Whye Teh**[1,2]

[1] Department of Statistics, University of Oxford, UK.
[2] DeepMind, UK.

## Abstract

Subsampling is used in convolutional neural networks (CNNs) in the form of pooling or strided convolutions, to reduce the spatial dimensions of feature maps and to allow the receptive fields to grow exponentially with depth. However, it is known that such subsampling operations are not translation equivariant, unlike convolutions that *are* translation equivariant. Here, we first introduce translation equivariant subsampling/upsampling layers that can be used to construct exact translation equivariant CNNs. We then generalise these layers beyond translations to general groups, thus proposing *group equivariant subsampling/upsampling*. We use these layers to construct group equivariant autoencoders (GAEs) that allow us to learn low-dimensional equivariant representations. We empirically verify on images that the representations are indeed equivariant to input translations and rotations, and thus generalise well to unseen positions and orientations. We further use GAEs in models that learn object-centric representations on multi-object datasets, and show improved data efficiency and decomposition compared to non-equivariant baselines.

## 1   Introduction

Convolutional Neural Networks (CNNs) are known to be more data efficient and show better generalisation on perceptual tasks than fully-connected networks, due to translation equivariance encoded in the convolutions: when the input image/feature map is translated, the output feature map also translates by the same amount. In typical CNNs, convolutions are used in conjunction with subsampling operations, in the form of pooling or strided convolutions, to reduce the spatial dimensions of feature maps and to allow receptive field to grow exponentially with depth. Subsampling/upsampling operations are especially necessary for convolutional autoencoders (ConvAEs) (Masci et al., 2011) because they allow efficient dimensionality reduction. However, it is known that subsampling operations implicit in strided convolutions or pooling layers are *not* translation equivariant (Zhang, 2019), hence CNNs that use these components are also not translation invariant. Therefore such CNNs and ConvAEs are not guaranteed to generalise to arbitrarily translated inputs despite their convolutional layers being translation equivariant.

Previous work, such as Zhang (2019); Chaman and Dokmanić (2020), has investigated how to enforce translation invariance on CNNs, but does not study equivariance with respect to symmetries beyond translations, such as rotations or reflections. In this work, we first describe subsampling/upsampling operations that preserve exact translation equivariance. The main idea is to sample feature maps on an input-dependent grid rather than a fixed one as in pooling or strided convolutions, and the grid is chosen according to a *sampling index* computed from the inputs (see Figure 1). Simply replacing the subsampling/upsampling in standard CNNs with such translation equivariant subsampling/upsampling operations leads to CNNs and transposed CNNs that can map between spatial inputs and low-dimensional representations in a translation equivariant manner.

---

*Corresponding author: <jin.xu@stats.ox.ac.uk>

35th Conference on Neural Information Processing Systems (NeurIPS 2021).

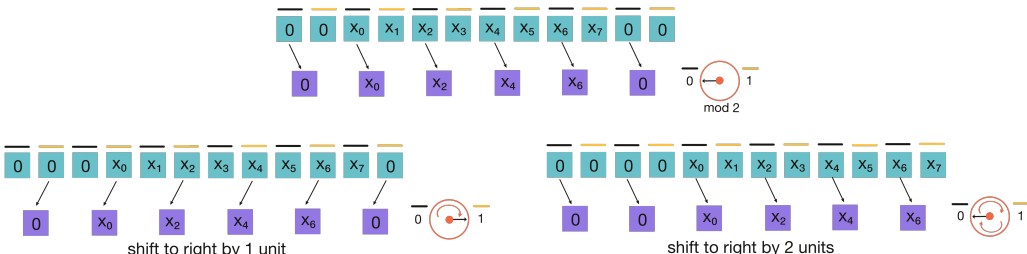

Figure 1: Equivariant subsampling on 1D feature maps with a scale factor $c = 2$. The input feature map has length 8, and initially we sample from odd positions determined by Equation (1) (top). When the original feature map is shifted to the right by 1 unit (bottom left), the sampling index becomes 1, so we instead sample from even positions. When the feature map is shifted to the right by 2 units (bottom right), we again sample from odd positions, but the outputs have been shifted to the right by 1 unit correspondingly.

We further generalise the proposed subsampling/upsampling operations from translations to arbitrary groups, proposing *group equivariant subsampling/upsampling*. In particular we identify subsampling as mapping features on groups $G$ to features on subgroups $K$ (vice versa for upsampling), and identify the sampling index as a coset in the quotient space $G/K$. See Appendix A for a primer on group theory that is needed to describe this generalisation. We note that group equivariant subsampling is different to *coset pooling* introduced in Cohen and Welling (2016), which instead gives features on the quotient space $G/K$, and discuss differences in detail in Section 4. Similar to the translation equivariant subsampling/upsampling, group equivariant subsampling/upsampling can be used with group equivariant convolutions to produce group equivariant CNNs. Using such group equvariant CNNs we can construct group equivariant autoencoders (GAEs) that separate representations into an invariant part and an equivariant part.

While there is a growing body of literature on group equivariant CNNs (G-CNNs) (Cohen and Welling, 2016, 2017; Worrall et al., 2017; Weiler et al., 2018b,a; Thomas et al., 2018; Weiler and Cesa, 2019a), such equivariant convolutions usually preserve the spatial dimensions of the inputs (or lift them to even higher dimensions) until the final invariant pooling layer. There is a lack of exploration on how to reduce the spatial dimensions of such feature maps while preserving exact equivariance, to produce low-dimensional equivariant representations. This work attempts to fill in this gap. Such low-dimensional equivariant representations can be employed in representation learning methods, allowing various advantages such as interpretability, out-of-distribution generalisation, and better sample complexity. When using such learned representations in downstream tasks such as abstract reasoning, reinforcement learning, video modelling, scene understanding, it is especially important for representations to be equivariant rather than invariant in these tasks, because transformations and how they act on feature spaces are critical information, rather than nuisance as in image classification problems.

In summary, we make the following contributions: (i) We propose subsampling/upsampling operations that preserve translational equivariance. (ii) We generalise the proposed subsampling/upsampling operations to arbitrary symmetry groups. (iii) We use equivariant subsampling/upsampling operations to construct GAEs that gives low-dimensional equivariant representations. (iv) We empirically show that representations learned by GAEs enjoys many advantages such as interpretability, out-of-distribution generalisation, and better sample complexity.

## 2 Equivariant Subsampling and Upsampling

### 2.1 Translation Equivariant Subsampling for CNNs

In this section we describe the proposed translation equivariant subsampling scheme for feature maps in standard CNNs. Later in Section 2.2, we describe how this can be generalised to group equivariant subsampling for feature maps on arbitrary groups.

**Standard subsampling** Feature maps in CNNs can be seen as functions defined on the integer grid, e.g. $\mathbb{Z}$ for 1D feature maps, and $\mathbb{Z}^2$ for 2D. Hence we represent feature maps as $f : \mathbb{Z} \to R^d$, where

$d$ is the number of feature map channels. For simplicity, we start with 1D and move on to the 2D case. Typically, subsampling in CNNs is implemented as either strided convolution or (max) pooling, and they can be decomposed as

$$\text{CONV}_k^c = \text{SUBSAMPLING}^c \circ \text{CONV}_k^1$$
$$\text{MAXPOOL}_k^c = \text{SUBSAMPLING}^c \circ \text{MAXPOOL}_k^1$$

where subscripts denote kernel sizes and superscripts indicate strides. $c \in \mathbb{N}$ is the scale factor for SUBSAMPLING, and this operation simply restricts the input domain of the feature map from $\mathbb{Z}$ to $c\mathbb{Z}$, without changing the corresponding function values.

**Translation equivariant subsampling**    In our equivariant subsampling scheme, we instead restrict the input domain to $c\mathbb{Z} + i$, the integers $\equiv i \mod c$, where $i$ is a sampling index determined by the input feature map. The key idea is to choose $i$ such that it is shifts by $t(\mod c)$ when the input is translated by $t$, to ensure that the same features are subsampled upon translation. Let $i$ be given by the mapping $\Phi_c : \mathcal{I}_{\mathbb{Z}} \to \mathbb{Z}/c\mathbb{Z}$. $\mathcal{I}_{\mathbb{Z}}$ denotes the space of vector functions on $\mathbb{Z}$ and $\mathbb{Z}/c\mathbb{Z}$ is the space of remainders upon division by $c$.

$$i = \Phi_c(f) = \text{mod}(\arg\max_{x \in \mathbb{Z}} \|f(x)\|_1, c) \tag{1}$$

where $\| \cdot \|_1$ denotes $L^1$-norm (other choices of norm are equally valid). Other choices for $\Phi_c$ are equally valid as long as they satisfy translation equivariance, ensuring that the same features are subsampled upon translation of the input:

$$\Phi_c(f(\cdot - t)) = \text{mod}(\Phi_c(f) + t, c). \tag{2}$$

Note that this holds for Equation (1) provided the argmax is unique, which we assume for now (see Appendix B.1 for a discussion of the non-unique case). We can decompose the subsampled feature map defined on $c\mathbb{Z} + i$ into its values and the offset index $i$, expressing it as $[f_b, i] \in (\mathcal{I}_{c\mathbb{Z}}, \mathbb{Z}/c\mathbb{Z})$, where $f_b$ is the translated output feature map such that $f_b(cx) = f(cx + i)$ for $x \in \mathbb{Z}$.

The subsampling operation described above, which maps from $\mathcal{I}_{\mathbb{Z}}$ to $(\mathcal{I}_{c\mathbb{Z}}, \mathbb{Z}/c\mathbb{Z})$ is translation equivariant: when the feature map $f$ is translated to the right by $t \in \mathbb{Z}$, one can verify that $f_b$ will be translated to the right by $c\lfloor \frac{i+t}{c} \rfloor$, and the sampling index for the translated inputs would become $\text{mod}(i + t, c)$. We provide an illustration for $c = 2$ in Figure 1, and describe formal statements and proofs later for the general cases in Section 2.2.

**Multi-layer case**    For the subsequent layers, the feature map $f_b$ is fed into the next convolution, and the sampling index $i$ is appended to a list of outputs. When the above translation equivariant subsampling scheme is interleaved with convolutions in this way, we obtain an exactly translation equivariant CNN, where each subsampling layer with scale factor $c_k$ produces a sampling index $i_k \in \mathbb{Z}/c_k\mathbb{Z}$. Hence the equivariant representation output by the CNN with $L$ subsampling layers is a final feature map $f_L$ and a $L$-tuple of sampling indices $(i_1, \ldots, i_L)$. This tuple can in fact be expressed equivalently as a single integer by treating the tuple as mixed radix notation and converting to decimal notation. We provide details of this multi-layer case in Appendix B.2, including a rigorous formulation and its equivariance properties.

**Translation equivariant upsampling**    As a counterpart to subsampling, upsampling operations increase the spatial dimensions of feature maps. We propose an equivariant upsampling operation that takes in a feature map $f \in \mathcal{I}_{c\mathbb{Z}}$ and a sampling index $i \in \mathbb{Z}/c\mathbb{Z}$, and outputs a feature map $f_u \in \mathcal{I}_{\mathbb{Z}}$, where we set $f_u(cx + i) = f(cx)$ and $\mathbf{0}$ everywhere else. This works well enough in practice, although in conventional upsampling the output feature map is often a smooth interpolation of the input feature map. To achieve this with equivariant upsampling, we can additionally apply average pooling with stride 1 and kernel size $> 1$.

**2D Translation equivariant subsampling**    When feature maps are $2D$, they can be represented as functions on $\mathbb{Z}^2$. The sampling index becomes a 2-element tuple given by:

$$(x^*, y^*) = \arg\max_{(x,y) \in \mathbb{Z}^2} \|f(x)\|_1$$
$$(i, j) = (\text{mod}(x^*, c), \text{mod}(y^*, c))$$

and we subsample feature maps by restricting the input domain to $c\mathbb{Z}^2 + (i, j)$. The multi-layer construction and upsampling is analogous to the 1D-case.

## 2.2 Group Equivariant Subsampling and Upsampling

In this section, we propose group equivariant subsampling by starting off with the 1D-translation case in Section 2.1, and provide intuition for how each component of this special case generalises to arbitrary discrete groups $G$. We then proceed to mathematically formulate group equivariant subsampling, and prove that it is indeed $G$-equivariant.

**Feature maps on groups** First recall that the feature maps for the 1D-translation case were defined as functions on $\mathbb{Z}$, or $f \in \mathcal{I}_{\mathbb{Z}}$ for short. To extend this to the general case, we consider feature maps $f$ as functions on a group $G$, i.e. $f \in \mathcal{I}_G = \{f : G \to V\}$[2] where $V$ is a vector space, as is done in e.g. group equivariant CNNs (G-CNNs) (Cohen and Welling, 2016). Note that translating feature maps $f$ on $\mathbb{Z}$ by displacement $u$ is effectively defining a new feature map $f'(\cdot) = f(\cdot - u)$. In the general case, we say that the group action on the feature space is given by

$$[\pi(u)f](g) = f(u^{-1}g) \tag{3}$$

where $\pi$ is a group representation describing how $u \in G$ acts on the feature space.

**Recap: translation equivariant subsampling** Recall that standard subsampling that occurs in pooling or strided convolutions for 1D translations amounts to restricting the domain of the feature map from $\mathbb{Z}$ to $c\mathbb{Z}$, whereas equivariant subsampling also produces a sampling index $i \in \mathbb{Z}/c\mathbb{Z}$, an integer mod $c$, and that this is equivalent to restricting the input domain to $c\mathbb{Z} + i$. $i$ is given by the translation equivariant mapping $\Phi_c : \mathcal{I}_{\mathbb{Z}} \to \mathbb{Z}/c\mathbb{Z}$. We can translate the input domain back to $c\mathbb{Z}$, and represent the output of subsampling as $[f_b, i] \in (\mathcal{I}_{c\mathbb{Z}}, \mathbb{Z}/c\mathbb{Z})$, where $f_b$ is the translated output feature map and $f_b(cx) = f(cx + i)$ for $x \in \mathbb{Z}$.

**Group equivariant subsampling** Similarly in the general case, for a feature map $f \in \mathcal{I}_G$, standard subsampling can be seen as restricting the domain from the group $G$ to a subgroup $K$, whereas equivariant subsampling additionally produces a sampling index $pK \in G/K$, where the quotient space $G/K = \{gK : g \in G\}$ is the set of (left) *cosets* of $K$ in $G$. Note that we have rewritten $i$ as $p$ to distinguish between the 1D translation case and the general group case. This is equivalent to restricting the $f$ to the coset $pK$. The choice of the coset $pK$ is given by equivariant map $\Phi : \mathcal{I}_G \to G/K$ (the action of $G$ on $G/K$ is given by $u(gK) = (ug)K$ for $u, g \in G$), such that $pK = \Phi(f)$. This restriction of $f$ to $pK$ can also be thought of as having an output feature map $f_b$ on $K$ and choosing a coset representative element $\bar{p} \in pK$, such that $f_b(k) = f(\bar{p}k)$. This choice of coset representative is described by a function $s : G/K \to G$, such that $\bar{p} = s(pK)$. The function $s$ is called a section and should satisfy $s(pK)K = pK$.

Now let us formulate subsampling and upsampling operations $S_b\!\downarrow_K^G$ and $S_u\!\uparrow_K^G$ mathematically and prove its $G$-equivariance. Let $\mathcal{I}_K = \{f : K \to V'\}$ be the space of feature map on $K$. $S_b\!\downarrow_K^G$ takes in a feature map $f \in \mathcal{I}_G$ and produces a feature map $f_b \in \mathcal{I}_K$ and a coset in $G/K$. In reverse, the upsampling operation $S_u\!\uparrow_K^G$ takes in a feature map in $\mathcal{I}_K$, a coset in $G/K$, and produces a feature map in $\mathcal{I}_G$. We use a section $s : G/K \to G$ to represent a coset with a representative element in $G$, and point out that equivariance holds for any choice of $s$.

Formally, given an equivariant map $\Phi : \mathcal{I}_G \to G/K$ (we will discuss how to construct such a map in Section 2.3), and a fixed section $s : G/K \to G$ such that $\bar{p} = s(pK)$, the subsampling operation $S_b\!\downarrow_K^G : \mathcal{I}_G \to \mathcal{I}_K \times G/K$ is defined as:

$$pK = \Phi(f), \quad f_b(k) = f(\bar{p}k) \text{ for } k \in K$$
$$[f_b, pK] = S_b\!\downarrow_K^G(f; \Phi), \tag{4}$$

while the upsampling operation $S_u\!\uparrow_K^G : \mathcal{I}_K \times G/K \to \mathcal{I}_G$ is defined as:

$$f_u(g) = f(\bar{p}^{-1}g) \text{ if } g \in K \text{ else } \mathbf{0}$$
$$f_u = S_u\!\uparrow_K^G(f, pK). \tag{5}$$

---

[2]This is not to be confused with the space of Mackey functions in, e.g., Cohen et al. (2019), and rather it is the space of unconstrained functions on $G$.

To make the output of the upsampling dense rather than sparse, one can apply arbitrary equivariant smoothing functions such as average pooling with stride 1 and kernel size $> 1$, to compensate for the fact that we extend with $\mathbf{0}$s rather than values close to their neighbours. In practice, we observe that upsampling without any smoothing function works well enough.

The statement on the equivariance of $S_b\!\downarrow_K^G$ and $S_u\!\uparrow_K^G$ requires we specify the action of $G$ on the space $\mathcal{I}_K \times G/K$, which we denote as $\pi'$. For any $u \in G$,

$$p'K = upK, \quad f'_b = \pi(\bar{p}'^{-1}u\bar{p})f_b$$
$$[f'_b,\ p'K] = \pi'(u)[f_b,\ pK] \tag{6}$$

**Lemma 2.1.** $\pi'$ *defines a valid group action of $G$ on the space $\mathcal{I}_K \times G/K$.*

We can now state the following equivariance property (See Appendix D for a proof):

**Proposition 2.2.** *If the action of group $G$ on the space $\mathcal{I}_G$ and $\mathcal{I}_K \times G/K$ are specified by $\pi, \pi'$ (as defined in Equations (3) and (6)), and $\Phi : \mathcal{I}_G \to G/K$ is an equivariant map, then the operations $S_b\!\downarrow_K^G$ and $S_u\!\uparrow_K^G$ as defined in Equations (4) and (5) are equivariant maps between $\mathcal{I}_G$ and $\mathcal{I}_K \times G/K$.*

In fact, we can also prove the converse (See Appendix D):

**Proposition 2.3.** *If $S_b\!\downarrow_K^G : \mathcal{I}_G \to \mathcal{I}_K \times G/K$ (as defined in Equation (4)) is an equivariant map, then the corresponding $\Phi : \mathcal{I}_G \to G/K$ must be equivariant.*

The above implies that $\Phi$ must depend on the input feature map $f$.

## 2.3 Constructing $\Phi$

We use the following simple construction of the equivariant mapping $\Phi : \mathcal{I}_G \to G/K$ for subsampling/upsampling operations, although any equivariant mapping would suffice. For an input feature map $f \in \mathcal{I}_G$, we define

$$pK = \Phi(f) := (\arg\max_{g \in G} \|f(g)\|_1)K \tag{7}$$

Provided that the argmax is unique, it is easy to show that $(up) \cdot K = \Phi(\pi(u)f)$, hence $\Phi$ is equivariant. In practice one can insert arbitrary equivariant layers to $f$ before and after we take the norm $\|\cdot\|_1$ to avoid a non-unique argmax (see Appendix F). Note that the argmax function alone may not be noise-robust. In Appendix E.2, we empirically show that applying smoothing equivariant layers before taking the argmax would improve the stability of the output sampling indices.

**Non-unique argmax case** When the input feature map $f \in \mathcal{I}_G$ has inherent symmetries, i.e. there exists $u \in G, u \neq e$, such that $f = \pi(u)f$, one cannot avoid a non-unique argmax in Equation (7). That is because if there is a unique argmax $g^*$ such that $g^* = \arg\max_{g \in G} \|f(g)\|_1$, we would have:

$$f(u^{-1}g^*) = f(g^*) = \max_{g \in G} \|f(g)\|_1$$

Therefore $u^{-1}g^*$ is also a valid argmax, hence the argmax is not unique. For symmetric inputs, the equivariant map $\Phi$ would give a set of sampling indices (cosets) rather than a single one. If we instead consider including this set of sampling indices in $z_{eq}$, and let group acts on this set, it can be shown that the exact equivariance would still hold. In practice, we uniformly sample a sampling index from this set to perform subsampling, and the subsampled feature maps will be the same for all sampling indices from this set because the inputs are symmetric. This complexity is unavoidable because an equivariant map that maps the feature map to a single coset does not exist in this case. However, perfectly symmetric inputs are very rare for real-world applications and we only encounter this problem for synthetic data.

## 3 Application: Group Equivariant Autoencoders

Group equivariant autoencoders (GAEs) are composed of alternating G-convolutional layers and equivariant subsampling/upsampling operations for the encoder/decoder. One important property

of GAEs is that the final subsampling layer of the encoder subsamples to a feature map defined on the trivial group $\{e\}$, outputting a vector (instead of a feature map) that is *invariant*. For the 1D-translation case, suppose the input to the final subsampling layer is a feature map $f$ defined on $\mathbb{Z}$. Then the final layer produces an invariant vector $f_b(0) = f(i_L)$ where $i_L = \arg\max_{x \in \mathbb{Z}} \|f(x)\|_1$. Note that there is no scale factor $c_L$ here. Intuitively we can think of this as setting the scale factor $c_L = \infty$. Hence the encoder of the GAE outputs a representation that is disentangled into an invariant part $z_{\text{inv}} = f_b(0)$ (the vector output by the final subsampling layer) and an equivariant part $z_{\text{eq}} = (i_1, ..., i_L)$.

For the general group case, instead of specifying scale factors as in Section 2.1, we specify a sequence of nested subgroups $G = G_0 \geq G_1 \geq \cdots \geq G_L = \{e\}$, where the feature map for layer $l$ is defined on subgroup $G_L$. For example, for the $p4$ group $G = \mathbb{Z} \rtimes \mathsf{C}_4$, we can use the following sequence for subsampling: $\mathbb{Z} \rtimes \mathsf{C}_4 \geq 2\mathbb{Z} \rtimes \mathsf{C}_4 \geq 4\mathbb{Z} \rtimes \mathsf{C}_4 \geq 8\mathbb{Z} \rtimes \mathsf{C}_2 \geq \{e\}$. Note that for the final two layers of this example, we are subsampling translations and rotations jointly.

We lift the input defined on the homogeneous input space to $\mathcal{I}_G$ (see Appendix A.3 for details on homogeneous spaces and lifting), and treat $f_0 \in \mathcal{I}_G$ as inputs to the autoencoders. The group equivariant encoder ENC can be described as follows:

$$[f_l,\ p_l G_l] = S_b\!\downarrow_{G_l}^{G_{l-1}}(\text{G-CNN}_{l-1}^E(f_{l-1}); \Phi_l)$$
$$[z_{\text{inv}}, z_{\text{eq}}] = [f_L(e),\ (p_1 G_1, p_2 G_2, \ldots, p_L G_L)] \tag{8}$$

where $l = 1, \ldots, L$ and G-CNN$_l(\cdot)$ denotes G-convolutional layers before the $l$th subsampling layer.

The decoder DEC simply goes in the opposite direction, and can be written formally as:

$$f_L \text{ is defined on } G_L = \{e\} \text{ and } f_L(e) = z_{\text{inv}}$$
$$f_{l-1} = \text{G-CNN}_{l-1}^D(S_u\!\uparrow_{G_l}^{G_{l-1}}(f_l,\ p_l G_l)) \tag{9}$$

where $l = L, \ldots, 1$ and $\hat{f} = f_0$ gives the final reconstruction.

Recall from Section 2.1 that the tuple $(i_1, \ldots, i_L)$ can be expressed equivalently as a single integer. Similarly, the tuple $(p_1 G_1, p_2 G_2, \ldots, p_L G_L)$ can be expressed as a single group element in $G$. We show in Appendix B.2 that the action implicitly defined on the tuple via Equation (6) simplifies elegantly to the left-action on the single group element in $G$.

We now have the following properties for the learned representations (see Appendix D for a proof):

**Proposition 3.1.** *When* ENC *and* DEC *are given by Equations* (8) *and* (9)*, and the group actions are specified as in Equation* (3) *and Equation* (6)*, for any $g \in G$ and $f \in \mathcal{I}_G$, we have*

$$[z_{inv}, g \cdot z_{eq}] = \text{ENC}(\pi(g)f)$$
$$\pi(g)\hat{f} = \text{DEC}(z_{inv}, g \cdot z_{eq})$$

## 4 Related Work

**Group equivariant neural networks**  The equivariant subsampling/upsampling that we propose deals with feature maps (functions) defined on the space of the group $G$ or its subgroups $K$, which transform under the *regular representation* with the group action. Hence our equivariant subsampling/upsampling is compatible with *lifting-based* group equivariant neural networks defined on discrete groups (Cohen and Welling, 2016; Hoogeboom et al., 2018; Romero and Hoogendoorn, 2020; Romero et al., 2020) that define a mapping between feature maps on $G$. We also discuss the extension of group equivariant subsampling to be compatible with those defined on continuous/Lie groups (Cohen et al., 2018a; Esteves et al., 2018; Finzi et al., 2020; Bekkers, 2020; Hutchinson et al., 2021) in Section 6. This is in contrast to group equivariant neural networks that do not use lifting and use *irreducible representations*, defining mappings between feature maps on the input space $\mathbf{X}$. (Cohen and Welling, 2017; Worrall et al., 2017; Thomas et al., 2018; Kondor et al., 2018; Weiler et al., 2018b,a; Weiler and Cesa, 2019a; Esteves et al., 2020; Fuchs et al., 2020).

**Coset pooling**  In particular, Cohen and Welling (2016) propose *coset pooling*, which is also a method for equivariant subsampling. Here a feature map $f$ on $G$ is mapped onto a feature map $\Phi(f)$ on $G/K$ (as opposed to $K$, for our equivariant subsampling) as follows:

$$\Phi(f)(gK) = \text{POOL}_{k \in K} f(gk) \tag{10}$$

such that the feature values on the coset $gK$ are pooled. For the 1D-translation case, where $G = \mathbb{Z}, K = c\mathbb{Z}$, this amounts to pooling over every $c$th pixel, which disrupts the locality of features as opposed to our equivariant subsampling that preserves locality, and hence is more suitable to use with convolutions for translation equivariance. See Figure 2 for a visual comparison. As such, the $p4$-CNNs in Cohen and Welling (2016) use standard max pooling with stride=2 rather than coset pooling for $\mathbb{Z}^2$, and coset-pooling is only used in the final layer to pool over feature maps across 90-degree rotations, achieving exact rotation equivariance but imperfect translation equivariance. In our work, we use translation equivariant subsampling in the earlier layers and rotation equivariant subsampling in the final layers to achieve exact roto-translation equivariance.

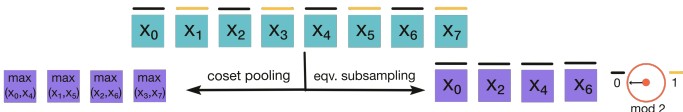

Figure 2: Coset (max) pooling vs. equivariant subsampling.

**Unsupervised disentangling and object discovery**    GAEs produce equivariant ($z_{\text{eq}}$) and invariant ($z_{\text{inv}}$) representations, effectively separating position and pose information with other semantic information. This relates to unsupervised disentangling (Higgins et al., 2017; Chen et al., 2018; Kim and Mnih, 2018; Zhao et al., 2017) where different factors of variation in the data are separated in different dimensions of a low-dimensional representation. However unlike equivariant subsampling, there is no guarantee of any equivariance in the low-dimensional representation, making the resulting disentangled representations less interpretable. Works on unsupervised object discovery (Burgess et al., 2019; Greff et al., 2019; Engelcke et al., 2020; Locatello et al., 2020) learn object-centric representations, and we showcase GAEs in MONet (Burgess et al., 2019) where we replace their VAE with a V-GAE in order to separate position and pose information and learn more interpretable representations of objects in a data-efficient manner.

**Shift-invariance in CNNs**    As early as Simoncelli et al. (1992), it has been discussed that shift-invariance cannot hold for conventional subsampling. Although standard subsampling operations such as pooling or strided convolutions are not *exactly* shift invariant, they do not prevent strong performance on classification tasks (Scherer et al., 2010). Nonetheless, Zhang (2019) integrates anti-aliasing to improve shift-invariance, showing that it leads to better performance and generalisation on classification tasks. Chaman and Dokmanić (2020) explore a similar strategy to our equivariant subsampling by partitioning feature maps into polyphase components and select the component with the highest norm. However, unlike the proposed group equivariant subsampling/upsampling which tackle general equivariance for arbitrary discrete groups, both works focus only on translation invariance.

**Equivariant/invariant autoencoders**    GAEs learn exact low-dimensional equivariant representations under the autoencoding framework, and this has also been explored in previous work. Locatello et al. (2019) constructs autoencoders that are equivariant to the $D_{12}$ group ($30°$ rotations and reflections) but not to translations, and the dimension of their learned representation grows with the group size. Lohit and Trivedi (2020) explores a rotation-invariant encoder on spheres with a global pooling layer and a rotation-invariant loss function, without inserting subsampling layers between convolutional layers. Moreover, unlike the work above that focuses on group equivariant neural networks, Hinton et al. (2011); Sabour et al. (2017); Kosiorek et al. (2019) learn equivariant representations with capsule networks. However, general capsule networks do not come guaranteed exact equivariances or invariances.

## 5 Experiments

In this section, we compare the performance of GAEs with equivariant subsampling to their non-equivariant counterparts that use standard subsampling/upsampling in object-centric representation learning. We show that GAEs give rise to more interpretable representations that show better sample complexity and generalisation than their non-equivariant counterparts. In Appendix E.1, we show that we can also observe generalisation performance gains when using group equivariant subsampling for classification tasks.

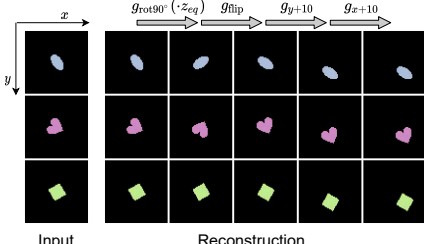
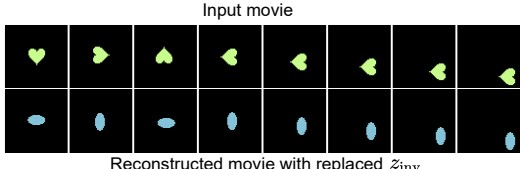

Figure 3: (Left) Manipulating reconstructions by modifying the equivariant part $z_{eq}$. The second column are the original reconstructions, which match the inputs well. The subsequent columns are reconstructions decoded from modified $z_{eq}$. We transform $z_{eq}$ with a sequence of group elements, and show the resulting reconstructions. (Right) Manipulating reconstruction shape by modifying $z_{inv}$.

**Models and Baselines**    (G-)Convolutional autoencoders (G)ConvAE are composed of alternating (G-)convolutional layers and subsampling/upsampling operations with a final MLPs applied to the flattened feature maps. We categorize models by the types of equivariance preserved by the convolutional layers. We consider three different discrete symmetry groups: $p1$ (only translations), $p4$ (composition of translations and 90 degree rotations), $p4m$ (composition of translations, 90 degree rotations and mirror reflection). The baseline models are: ConvAE-$p1$ (standard convolutional autoencoders), GConvAE-$p4$, GConvAE-$p4m$, where the corresponding equivariance is preserved in the (G-)convolutional layers but not in the subsampling/upsampling operations. The equivariant counterparts of these baseline models are GAE-$p1$, GAE-$p4$, GAE-$p4m$, where the subsampling/upsampling operations are also equivariant. For baseline models, we use a scale factor of 2 for all subsampling/upsampling layers. For GAEs, we subsample first the translations, then rotations, followed by reflections, all with scale factor 2. e.g. for GAE-$p4m$, the feature maps at each layer are defined on the following chain of nested subgroups: $\mathbb{Z}^2 \rtimes (\mathsf{C}_4 \rtimes \mathsf{C}_2) \geq (2\mathbb{Z})^2 \rtimes (\mathsf{C}_4 \rtimes \mathsf{C}_2) \geq (4\mathbb{Z})^2 \rtimes (\mathsf{C}_4 \rtimes \mathsf{C}_2) \geq (8\mathbb{Z})^2 \rtimes (\mathsf{C}_4 \rtimes \mathsf{C}_2) \geq (16\mathbb{Z})^2 \rtimes (\mathsf{C}_2 \rtimes \mathsf{C}_2) \geq \{e\}$. As in Cohen and Welling (2016), we rescale the number of channels such that the total number of parameters of these models roughly match each other.

**Data**    To demonstrate basic properties of GAEs and compare sample complexity under the single object scenario, we use Colored-dSprite (Matthey et al., 2017) and a modification of FashionMNIST (Xiao et al., 2017), where we first apply zero-padding to reach a size of $64 \times 64$, followed by random shifts, rotations and coloring. For multi-object datasets, we use Multi-dSprites (Kabra et al., 2019) and CLEVR6 which is a variant of CLEVR (Johnson et al., 2017) with up to 6 objects. All input images are resized to a resolution of $64 \times 64$.

See Appendix F and our reference implementation [3] for more details on hyperparameters and data preprocessing. Our implementation is built upon open source projects Harris et al. (2020); Paszke et al. (2019); Yadan (2019); Weiler and Cesa (2019b); Engelcke et al. (2020); Hunter (2007); Waskom (2021).

### 5.1    Basic Properties: Equivariance, Disentanglement and Out-of-Distribution Generalization

**Equivariance**    The encoder-decoder pipeline in GAEs is exactly equivariant. In Figure 3, we train GAE-$p4m$ on 6400 examples from Colored-dSprites, and we show how to manipulate reconstructions by manipulating the equivariant representation $z_{eq}$ (left). If an image $x$ is encoded into $[z_{inv}, z_{eq}]$, then decoding $[z_{inv}, g \cdot z_{eq}]$ will give $g \cdot \hat{x}$ where $\hat{x}$ is the reconstruction of $x$. When the input has perfect symmetries (e.g. squares, ellipses in Figure 3), $z_{eq}$ is obtained by sampling from a set of sampling indices but different sampling indices in this set would give the same reconstruction (see Section 2.3).

**Disentanglement**    The learned representations in GAEs are disentangled into an invariant part $z_{inv}$ and an equivariant part $z_{eq}$. In Figure 3 (left), we vary the equivariant part while the invariant part remains the same. In Figure 3 (right), we show the frames of a movie of a heart, and show its reconstruction after replacing $z_{inv}$ representing a heart with that of an ellipse. Note that the ellipse shape undergoes the same sequence of transformations as the heart.

---

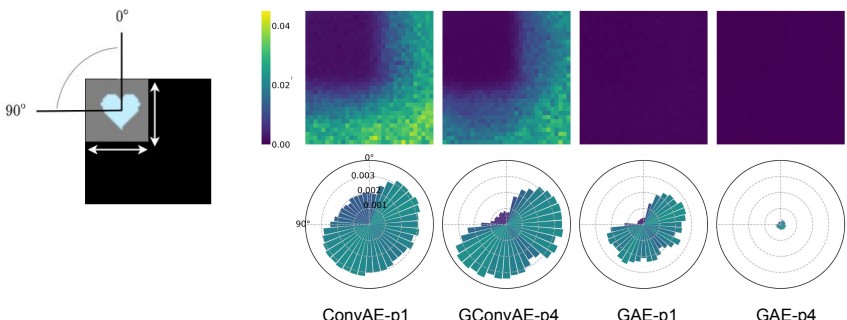

Figure 4: Generalisation to out-of-distribution object locations and poses. During training, we constrain shapes to be in the top-left quarter, and the orientation to be always less than 90 degrees. On the right, we compare the error of reconstructions of different models generalise on objects at unseen locations in the first row, and how they generalise to unseen orientations in the second row.

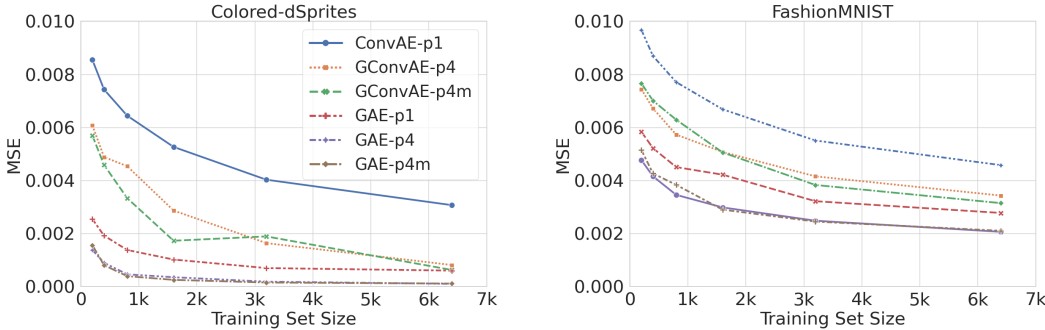

Figure 5: Reconstruction error on single object datasets

**Out-of-distribution generalisation** GAEs can generalise to data with unseen object locations and poses. We train an GAE-$p4$ on 6400 constrained training examples, where we only use examples with locations in the top-left quarter and orientations within $[0, 90]$ degrees, as shown in Figure 4. During test time, we evaluate mean squared error (MSE) of reconstructions on unfiltered test data to see how models generalise to unseen location and poses. Both ConvAE-$p1$ and GConvAE-$p4$ cannot generalise well to object poses out of their training distribution. In contrast, GAE-$p1$ generalise to any locations without performance degradation but not to unseen orientations, while GAE-$p4$, which encodes both translation and rotation equivariance, generalises well to all locations and orientations. We only use heart shapes for evaluation, because the square and ellipse have inherent symmetries (see Section 2.3).

## 5.2   Single Object

Since GAEs are fully equivariant and can generalize to unseen object poses, it is natural to conjecture that such models can significantly improve data efficiency when symmetry-transformed data points are also plausible samples from the data distribution. We test this hypothesis on Colored-dSprites and transformed FashionMNIST, and the results are shown in Figure 5. On both datasets, equivariant autoencoders significantly outperform their non-equivariant counterparts for all considered training set sizes. In fact, as shown in the figure, equivariant models trained with a smaller training set size is often comparable to baseline models trained on a larger training set. Furthermore, the results demonstrate that it is beneficial to consider symmetries beyond translations in these problems: for both non-equivariant and equivariant models, variants that encode rotation and reflection symmetries consistently show better performance compared to models that only consider the translation symmetry.

## 5.3   Multiple Objects

In multi-object scenes, it is often more interesting to consider local symmetries associated with objects rather than the global symmetry for the whole image. To exploit object symmetries in

Table 1: Reconstruction error MSE ($\times 10^{-3}$) (mean(stddev) across 5 seeds) on multi-object datasets

| Dataset | Multi-dSprites | | | CLEVR6 | | |
|---|---|---|---|---|---|---|
| Training Set Size | 3200 | 6400 | 12800 | 3200 | 6400 | 12800 |
| MONet | 2.661(0.382) | 1.385(0.235) | 0.326(0.076) | 0.673(0.059) | 0.562(0.057) | 0.546(0.056)[1] |
| MONet-GAE-$p1$ | 0.659(0.103) | 0.359(0.025) | 0.264(0.042) | 0.473(0.064) | 0.432(0.052) | 0.388(0.016) |
| MONet-GAE-$p4$ | 0.563(0.195) | 0.317(0.060) | 0.231(0.067) | 0.461(0.025) | 0.414(0.022) | 0.413(0.018) |

Table 2: Foreground segmentation performance in terms of ARI (mean(stddev) across 5 seeds)

| Dataset | Multi-dSprites | | | CLEVR6 | | |
|---|---|---|---|---|---|---|
| Training Set Size | 3200 | 6400 | 12800 | 3200 | 6400 | 12800 |
| MONet | 0.597(0.022) | 0.747(0.049) | 0.891(0.009) | 0.829(0.055) | 0.878(0.023) | 0.865(0.033)[1] |
| MONet-GAE-$p1$ | 0.762(0.049) | 0.823(0.042) | 0.889(0.013) | 0.921(0.015) | 0.917(0.032) | 0.920(0.025) |
| MONet-GAE-$p4$ | 0.753(0.089) | 0.833(0.072) | 0.902(0.025) | 0.878(0.055) | 0.914(0.012) | 0.910(0.011) |

[1] *We excluded 2 outliers here as the baseline MONet occasionally fails during late-phase training.*

image data, one needs to first discover objects and separate them from the background, which is a challenging problem on its own. Currently, GAEs do not have inherent capability to solve these problems. In order to investigate whether our models could improve data efficiency in multi-object settings, we rely on recent work on unsupervised object discovery and only use GAEs to model object components. More specifically, we explored replacing component VAEs in MONet (Burgess et al., 2019) with V-GAEs (probabilistic version of our GAEs, where a standard Gaussian prior is put on $z_{\text{inv}}$ and $z_{\text{eq}}$ remains deterministic), and train models end-to-end. Again we study the low data regime to show results on data efficiency.

We train models on Multi-dSprites and CLEVR6 with training set sizes 3200, 6400 and 12800. We consider two evaluation metrics: mean squared error (MSE) to measure the overall reconstruction quality, and adjusted rand index (ARI), which is a clustering similarity measure ranging from 0 (random) to 1 (perfect) to measure object segmentation. As in Burgess et al. (2019), we only use foreground pixels to compute ARI. Component VAEs in MONet use spatial broadcast decoders (Watters et al., 2019) that broadcast the latent representation to a full scale feature map before feeding them into the decoders, and the decoders therefore do not need upsampling. It has the implicit effect of encouraging the smoothness of the decoder outputs. To encourage similar behaviour, we add average pooling layers with stride 1 and kernel size 3 to our equivariant decoders. As shown in Table 1, using GAEs to model object components significantly improves reconstruction quality, which is consistent with our findings in single-object scenario. As shown in Table 2, using GAEs to model object components also leads to better object discovery in the low data regimes, but this advantage seems to diminish as the dataset becomes sufficiently large.

## 6    Conclusions, Limitations and Future Work

**Conclusions**    We have proposed subsampling/upsampling operations that *exactly* preserve translation equivariance, and generalised them to define *exact* group equivariant subsampling/upsampling for discrete groups. We have used these layers in GAEs that allow learning low-dimensional representations that can be used to reliably manipulate pose and position of objects, and further showed how GAEs can be used to improve data efficiency in multi-object representation learning models.

**Limitations and Future work**    Although the equivariance properties of subsampling layers also hold for Lie groups, we have not discussed the practical complexities that arise with the continuous case, where feature maps are only defined on a finite subset of the group rather than the whole group. We leave this as important future work, as well as application of equivariant subsampling for tasks other than representation learning where equivariance/invariance is desirable e.g. object classification, localization (See Appendix E.1 for a preliminary exploration of classification tasks). Another limitation is that our work focuses on global equivariance, like most other works in the literature. An important direction is to extend to the case of local equivariances e.g. object-specific symmetries for multi-object scenes.

## Acknowledgments and Disclosure of Funding

We would like to thank Adam R. Kosiorek for valuable discussion. We also thank Lewis Smith, Desi Ivanova, Sheheryar Zaidi, Neil Band, Fabian Fuchs, Ning Miao, and Matthew Willetts for providing feedback on earlier versions of the paper, and anonymous reviewers for their constructive suggestions during the review process. JX gratefully acknowledges funding from Tencent AI Labs through the Oxford-Tencent Collaboration on Large Scale Machine Learning.

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
