# A Preliminaries

## A.1 Group, Coset and Quotient Space

A *group* $G$ is a set of elements equipped with a binary operation (denoted as $\cdot$) that satisfies the following group axioms:

1. (Closure) For all $a, b \in G$, $a \cdot b \in G$.
2. (Associative) For all $a, b, c \in G$, $(a \cdot b) \cdot c = a \cdot (b \cdot c)$.
3. (Identity element) There exists an identity element $e$ in $G$ such that, for any $a \in G$ we have $e \cdot a = a \cdot e = a$.
4. (Inverse element) For each $a \in G$, there exists an element $b \in G$ such that $a \cdot b = b \cdot a = e$ where $e$ is the identity element.

The centered dot $\cdot$ can sometimes be omitted if there is no ambiguity.

In this work, we are mainly interested in symmetry groups where each group element is associated with a symmetry of a pattern, which is a transformation that leaves the pattern invariant. In symmetry groups, the binary operation corresponds to composition of transformations.

A subset $H$ contained within $G$ is a *subgroup* of $G$ if it forms a group on its own under the same binary operation. Given a subgroup $H$ and an arbitrary group element $g \in G$, one can define *left cosets* of $H$ as follows:

$$gH = \{g \cdot h \mid h \in H\}$$

The left cosets of $H$ form a partition of $G$ for any choice of $H$, i.e. the union of all cosets is $G$ and all cosets defined above are either identical or have empty interception. The set of all left cosets is called the *quotient space* and is denoted as $G/H = \{gH \mid g \in G\}$.

As an example, all integers $\mathbb{Z}$ under addition forms a group and all multiples of $n$, denoted as $n\mathbb{Z}$ is a subgroup of $\mathbb{Z}$. For any integer $k \in \mathbb{Z}$, the set $n\mathbb{Z} + k$ containing all integers that has the remainder as $k$ divided by $n$, is a coset of $n\mathbb{Z}$. There are $n$ distinct cosets like this, and they form the quotient space $\mathbb{Z}/n\mathbb{Z}$.

## A.2 Group Homomorphism, Group Actions and Group Equivariance

Given two groups $(G, \cdot_G)$ and $(H, \cdot_H)$, a *group homomorphism* from $G$ to $H$ is function $f : G \to H$ such that for any $u, v \in G$

$$f(u \cdot_G v) = f(u) \cdot_H f(v).$$

It is a special mapping between two groups that is compatible with group structures. If $f$ is an one-to-one mapping, we call it a *group isomorphism*. Two groups $G_1$ and $G_2$ are isomporphic if there is an isomorphism between them, and this is written as $G_1 \cong G_2$.

A *group action* is a group homomorphism from a given group $G$ to the group of transformations on a space $\mathbf{X}$. We say the group $G$ acts on the space $\mathbf{X}$ and the transformation corresponding to $g \in G$ is a bijection on $\mathbf{X}$ that maps $x$ to $g \cdot x$.

If the group actions of $G$ on spaces $\mathbf{X}$ and $\mathbf{Y}$ are both defined, a function $f : \mathbf{X} \to \mathbf{Y}$ is said to be *group equivariant* if

$$g \cdot f(x) = f(g \cdot x)$$

## A.3 Homogeneous Spaces and Lifting Feature Maps

If the action of a group $G$ on the space $\mathbf{X}$ is defined, and the action is transitive (i.e. $\forall x, x' \in \mathbf{X}, \exists g \in G$, s.t. $x' = g \cdot x$), we refer to $\mathbf{X}$ as being a homogeneous space for $G$. There is a natural one-to-one correspondence between the homogeneous space $\mathbf{X}$ and disjoint subsets of the group $G$. Given an arbitrary origin $x_0 \in \mathbf{X}$, $H = \{g \in G | g \cdot x_0 = x_0\}$ is a subgroup of $G$, where $H$ is called the stabiliser of the origin. Because the group action on $\mathbf{X}$ is transitive, every element $x \in \mathbf{X}$ corresponds to a left coset in $s(x; x_0)H \in G/H$, where $s(x; x_0)$ is (any) group element that transforms $x_0$ to $x$. It can be shown that for $x, x' \in \mathbf{X}, x \neq x'$, $s(x; x_0)$ and $s(x'; x_0)$ are disjoint.

Because spatial data is often represented as functions on the homogeneous space $f_\mathbf{X} : x_i \mapsto f_i$, while lifting-based group equivariant neural networks operate on feature maps defined on the group, there is usually an operation called *lifting*, that maps the data to the feature space of functions on the group, before applying equivariant modules. Using the correspondence between $\mathbf{X}$ and the quotient space $G/H$, we can map each pair $(x_i, f_i)$ to the set $\{(g, f_i) | g \in s(x_i; x_0)H\}$. It can be seen as lifting the input feature map $f_\mathbf{X} : x_i \mapsto f_i$ to the feature map $\text{LIFT}(f_\mathbf{X}) : g \mapsto f_i$ for $g \in s(x_i; x_0)H$. In this work, we assume all input feature maps have been lifted to feature maps on the group.

### A.4 Wallpaper Groups

Wallpaper groups categorise symmetries of repetitive patterns on a 2D plane. For simplicity, we only considered 3 different types of wallpaper symmetry groups $p1$, $p4$, and $p4m$ in this work following Cohen and Welling (2016). These groups are named using the crystallographic notation, where $p$ standards for primitive cells, the next digit indicates the highest order of rotational symmetries, and $m$ stands for mirror reflection. All symmetries contained in these groups can be deduced from their name:

- $p1$: All 2D integer translations.
- $p4$: All compositions of 2D integer translations and rotations by a multiple of 90 degrees.
- $p4m$: All compositions of elements in $p4$ and the mirror reflection.

All three groups $p1$, $p4$, and $p4m$ can be constructed from basic additive groups of integers $\mathbb{Z}$ and cyclic groups $\mathsf{C}_n$ using the inner semi-direct product. Given a group $G$ with a normal subgroup $N$ (i.e. $\forall n \in N, g \in G, gng^{-1} \in N$), a subgroup $H$ (not necessarily a normal subgroup), if $G$ is the product of subgroups $G = NH = \{nh | n \in N, h \in H\}$, and $N \cap H = \{e\}$, we say $G$ is a inner semi-direct product of $N$ and $H$, written as $G = N \rtimes H$. Using semi-direct product, $p1$, $p4$, and $p4m$ can be expressed as:

$$p1 \cong \mathbb{Z}^2$$
$$p4 \cong \mathbb{Z}^2 \rtimes \mathsf{C}_4$$
$$p4m \cong \mathbb{Z}^2 \rtimes (\mathsf{C}_4 \rtimes \mathsf{C}_2) \tag{11}$$

If $G \cong N \rtimes H$, the binary and inverse operations for $G$ can be determined from its subgroups $N$ and $H$. We represent group elements in $G$ as a tuple $(n, h)$ where $n \in N$ and $h \in H$. Let $\phi_h(n) = hnh^{-1}$, the binary operation on $G$ can be given by:

$$(n_1, h_1) \cdot (n_2, h_2) = (n_1 \phi_{h_1}(n_2), h_1 h_2)$$

and the inverse for element in $G$ can also be derived from the above:

$$(n, h)^{-1} = (\phi_{h^{-1}}(n^{-1}), h^{-1})$$

These properties can be used to simplify the implementation of the considered groups $p1$, $p4$, and $p4m$ following the decomposition in Equation (11), and the operations for basic groups $\mathbb{Z}$ and $\mathsf{C}_n$ are easy to implement.

### A.5 Feature Maps in G-CNNs

A general mathematical framework is introduced in Cohen et al. (2019) to specify convolutional feature spaces used in G-CNNs, and feature maps are treated as fields over a homogeneous space. It covers most previous works on equivariant neural networks including Cohen and Welling (2016, 2017); Cohen et al. (2018b); Weiler and Cesa (2019a). Under this framework, one way to represent fields is through constrained functions defined on the whole symmetry group, also known as Mackey functions (Cohen et al., 2019).

Formally, let $G$ be a symmetry group, and $H \leq G$ together with $G$ determines the homogeneous space $G/H$. For a group representation $(\rho, V)$ of $H$, the action of the whole group $G$ on fields can be described by an induced representation $\pi = \text{Ind}_H^G \rho$, whose realisation depends on how we represent these fields. Below we specify the feature space $\mathcal{I}_M$ [4] for the Mackey function field representation

---
[4]$\mathcal{I}_M$ corresponds to $\mathcal{I}_G$ in Cohen et al. (2019)

discussed in (Cohen et al., 2018c, 2019):

$$\mathcal{I}_M = \{f : G \to V | f(gh) = \rho(h^{-1})f(g), \forall g \in G, h \in H\} \tag{12}$$

which forms a vector space. Moreover, when $\rho$ is a *regular representation*, which is the implicit choice of Gens and Domingos (2014); Kanazawa et al. (2014); Dieleman et al. (2015, 2016); Cohen and Welling (2016); Marcos et al. (2016), fields can also be represented as unconstrained functions on $G$ and the feature space can be written as

$$\mathcal{I}_G = \{f : G \to V'\}$$

with $V'$ being a different vector space from $V$. If $\rho$ is a regular representation.

Feature maps are represented as functions on $G$ in both $\mathcal{I}_M$ and $\mathcal{I}_G$, even though $\mathcal{I}_M$ have additional conditions given in Equation (12). Moreover, the induced representation $\pi = \mathrm{Ind}_H^G \rho$ for them have the same form:

$$[\pi(u)f](g) = f(u^{-1}g)$$

## B    Equivariant Subsampling and Upsampling

### B.1    Constructing $\Phi$

In Section 2.3, we provide a simple construction of the equivariant map $\Phi : \mathcal{I}_G \to G/K$ which gives the sampling indexes. The construction is a valid one if the argmax is unique. In practice one can insert arbitrary equivariant layers to $f$ before and after we take the norm $\|\cdot\|_1$ to avoid a non-unique argmax (see Appendix F). However, in theory, there could be cases that the argmax is always non-unique. We discuss this case below and provide a more complex construction for it.

One cannot avoid a non-unique argmax in Equation (7) when the input feature map $f \in \mathcal{I}_G$ has inherent symmetries, i.e. there exists $u \in G, u \neq e$, such that $f = \pi(u)f$. Assuming there is a unique argmax $g^*$ such that $g^* = \arg\max_{g \in G} \|f(g)\|_1$, we would have:

$$f(u \cdot g^*) = f(g^*) = \max_{g \in G} \|f(g)\|_1$$

Therefore $u \cdot g^*$ is also a valid argmax, hence the argmax is not unique. For example, when $f$ is a feature map representing a center-aligned circle, we would have $f = \pi(u)f$, where $u \in O(2)$ is associated with an arbitrary rotation around the center. One cannot find a unique argmax $g^*$ for this example, because the feature map would take the same function values at $u \cdot g^*$.

Under the circumstance described above, the argmax operation would return a set of elements where each one attains the function's largest values. We denote it as $S^* = \arg\max_{g \in G} \|f(g)\|_1$, where $S^*$ is a subset of $G$. To obtain the sampling index (a coset) $pK$, we sample uniformly from the set $S^*$, and let $\Phi$ outputs $pK$ where $p \sim S^*$. In this case, the map $\Phi$ is still equivariant in distribution even though it is now a stochastic map.

Note that it is possible to consider more sophisticated solutions or even use learnable modules for $\Phi$, which we leave for future work.

### B.2    Multiple Subsampling Layers

**Translation equivariant subsampling**    We can stack convolutional and translation equivariant subsampling layers to construct exactly translation equivariant CNNs. Unlike standard CNNs, each translation equivariant subsampling layer with a scale factor $c_k$ outputs a subsampling index $i_k$ in addition to the feature maps. Hence the equivariant representation output by the CNN with $L$ subsampling layers is a final feature map $f_L$ and a $L$-tuple of sampling indices $(i_1, ..., i_L)$.

In the multi-layer case, the $l$-th subsampling layer takes in a feature map $f$ on $\prod_{k=1}^{l-1} c_k \mathbb{Z}$ and outputs: 1) a feature map on $\prod_{k=1}^{l} c_k \mathbb{Z}$ and 2) a subsampling index $i_l \in \prod_{k=1}^{l-1} c_k \mathbb{Z} / \prod_{k=1}^{l} c_k \mathbb{Z} \cong \mathbb{Z}/c_l\mathbb{Z}$ given by:

$$(\prod_{k=1}^{l-1} c_k) \cdot i_l = p_l = \Phi_c(f) = \mathrm{mod}(\arg\max_{x \in (\prod_{k=1}^{l-1} c_k)\mathbb{Z}} \|f(x)\|_1, \prod_{k=1}^{l} c_k)$$

This is equivalent to treating the input feature map $f$ as a feature map $f'$ defined on $\mathbb{Z}$ (i.e. mapping the support of $f$ from $\prod_{k=1}^{l-1} c_k \mathbb{Z}$ to $\mathbb{Z}$ via division by $\prod_{k=1}^{l-1} c_k$), and the subsampling layer outputting: 1) a feature map on $c_l \mathbb{Z}$ and 2) a subsampling index $i_l \in \mathbb{Z}/c_l \mathbb{Z}$ given by:

$$i_l = \text{mod}(\arg\max_{x \in \mathbb{Z}} \|f'(x)\|_1, c_l)$$

Hence the tuple $(i_1, ..., i_L)$ that contains the sampling indices of all layers can be expressed equivalently as a single integer:

$$r_{\text{eq}} = \sum_{l=1}^{L} p_l = \sum_{l=1}^{L} (\prod_{k=1}^{l-1} c_k) \cdot i_l$$

where $r_{\text{eq}} \in \mathbb{Z}/(\prod_{k=1}^{L} c_k)\mathbb{Z}$. Note that the conversion between $r_{\text{eq}}$ and $(i_1, ..., i_L)$ can be seen as the conversion between *mixed radix notation* and decimal notation. Mixed radix notation is a mixed base numeral system where the numerical base varies from position to position, as opposed to base-n systems that have the same base for all positions[5]. Thus there is an one-to-one correspondence between the two. Moreover, when the input feature map is translated to the right by $t \in \mathbb{Z}$, $r_{\text{eq}}$ would become $\text{mod}(r_{\text{eq}} + t, \prod_{k=1}^{L} c_k)$. See the statement of this result for the general group case in Proposition B.1 and its proof in Appendix D.4.

**Group equivariant subsampling**  Similarly, given an input feature map $f \in \mathcal{I}_G$, we can construct CNNs/G-CNNs with multiple equivariant subsampling layers by specifying a sequence of nested subgroups $G = G_0 \geq G_1 \geq \cdots \geq G_L$. The $l$-th subsampling layer takes in a feature map on $G_{l-1}$, outputs a feature map on $G_l$ and a sampling index $p_l G_l \in G_{l-1}/G_l$. Formally, the $l$-th subsampling layer can be written as:

$$S_b\downarrow_{G_l}^{G_{l-1}} : \mathcal{I}_{G_{l-1}} \to \mathcal{I}_{G_l} \times G_{l-1}/G_l$$

The equivariant representation output by the CNNs/G-CNNs with $L$ subsampling layers is a feature map in $f_L \in G_L$ and a $L$-tuple $(p_1 G_1, p_2 G_2, \ldots, p_L G_L)$.

Similar to the 1D translation case, the sampling index tuple $(p_1 G_1, p_2 G_2, \ldots, p_L G_L)$ can be expressed equivalently as a single element in the quotient space $G/G_L$:

$$r_{\text{eq}} = (\bar{p}_1 \bar{p}_2 \ldots \bar{p}_L) G_L = \nu(p_1 G_1, p_2 G_2, \ldots, p_L G_L) \tag{13}$$

where $\bar{p}_l$ denote the coset representative for the quotient space $G_{l-1}/G_l$. $\nu$ is a bijection from $r_{\text{eq}}$ to the tuple, whose inverse can be computed by the following recursive procedure:

$$p_1' G_L = r_{\text{eq}}$$
$$p_l' = \bar{p}_{l-1}'^{-1} \cdot p_{l-1}'$$
$$(p_1' G_1, p_2' G_2, \ldots, p_L' G_L) = \nu^{-1}(r_{\text{eq}}) \tag{14}$$

**Proposition B.1.** *$\nu^{-1}$ is the inverse of $\nu$, hence $\nu$ is bijective. And $\forall u \in G$ we have:*

$$u \cdot \nu(p_1 G_1, p_2 G_2, \ldots, p_L G_L) = \nu(u \cdot (p_1 G_1, p_2 G_2, \ldots, p_L G_L)).$$

## C  Group Equivariant Autoencoders

In Appendix B.2 we discussed that we can stack multiple subsampling layers by specifying a sequence of nested groups $G = G_0 \geq G_1 \geq \cdots \geq G_L$, and the CNN/G-CNNs with $L$ subsampling layers would produce a feature map on $G_L$ and a tuple $z_{\text{eq}} = (p_1 G_1, p_2 G_2, \ldots, p_L G_L)$. Furthermore, we know from Proposition B.1 that there is an one-to-one correspondence between the tuple representation $z_{\text{eq}}$ and the single group element representation $r_{\text{eq}} = \nu(z_{\text{eq}}) \in G/G_L$. For group equivariant autoencoders, we specify a sequence of subgroups but with $G_L = \{e\}$. In this case, $r_{\text{eq}}$ would simply become a group element in $G$. And the group action simplifies to left-multiplying the corresponding group elements.

---

[5]A commonly used example of mixed radix notation is to express time, where e.g. 12:34:56 has a base of 24 for the hour digit, base 60 for the minute digit and base 60 for the second digit.

Although one can simply use the tuple output by the encoder to perform upsampling in the decoder (and hence use the same sequence of nested subgroups), this is not strictly necessary as one can use a different sequence of nested subgroups for the decoder and obtain the tuple using the decomposition procedure in Equation (14). Moreover, for more efficient implementation of GAEs, one does not need to pass through $\Phi$ in Equation (7) for every subsampling layer. It would suffice to obtain the tuple of subsampling indexes from the first subsampling layer using:

$$(p_1 G_1, p_2 G_2, \ldots, p_L G_L) = \nu^{-1}(\arg \max_{g \in G} \|f(g)\|_1) \tag{15}$$

## D  Proofs

### D.1  Proof of Lemma 2.1 (See page 5)

**Lemma 2.1.** $\pi'$ *defines a valid group action of $G$ on the space $\mathcal{I}_K \times G/K$.*

*Proof.* Since $\bar{p}$ and $\overline{up}$ are coset representatives for $pK$ and $(up)K$, we can let $p = \bar{p}k_p$, $up = \overline{up}k_{up}$, where $k_p, k_{up} \in K$. From Equation (6), note that $\bar{p}' = \overline{up} = upk_{up}^{-1}$. Hence

$$\begin{aligned} \bar{p}'^{-1}u\bar{p} &= (upk_{up}^{-1})^{-1}u(pk_p^{-1}) \\ &= k_{up}p^{-1}u^{-1}upk_p^{-1} \\ &= k_{up}k_p^{-1} \in K \end{aligned} \tag{16}$$

Hence $\pi'(u)$ (as defined in Equation (6)) defines a transformation from the space $\mathcal{I}_K \times G/K$ to itself.

To prove $\pi'$ is a group action, we would like to show that for all $u, u' \in G$

$$\pi'(u')(\pi'(u)[f_b,\ pK]) = \pi'(u'u)[f_b,\ pK]$$

Let $[f_b',\ p'K] = \pi'(u)[f_b,\ pK]$ and $[f_b'',\ p''K] = \pi'(u'u)[f_b,\ pK]$, by the definition of $\pi'$ in Equation (6), we have

$$p''K = ((u'u)p)K = u'(upK) = u'(p'K)$$

and

$$f_b'' = \pi(\bar{p}''^{-1}(u'u)\bar{p})f_b = \pi(\bar{p}''^{-1}u'\bar{p}')\pi(\bar{p}'^{-1}u\bar{p})f_b = \pi(\bar{p}''^{-1}u'\bar{p}')f_b'.$$

Hence

$$[f_b'',\ p''K] = \pi'(u')[f_b',\ p'K]$$

It is easy to also check that

$$[f_b,\ pK] = \pi'(e)[f_b,\ pK]$$

Therefore $\pi'$ defines a valid group action. $\qquad\square$

### D.2  Proof of Proposition 2.2 (See page 5)

**Proposition 2.2.** *If the action of group $G$ on the space $\mathcal{I}_G$ and $\mathcal{I}_K \times G/K$ are specified by $\pi, \pi'$ (as defined in Equations (3) and (6)), and $\Phi : \mathcal{I}_G \to G/K$ is an equivariant map, then the operations $S_b\downarrow_K^G$ and $S_u\uparrow_K^G$ as defined in Equations (4) and (5) are equivariant maps between $\mathcal{I}_G$ and $\mathcal{I}_K \times G/K$.*

*Proof.* We first define a *restrict* operation on $f \in \mathcal{I}_G$ and an *extend* operation on $f_1 \in \mathcal{I}_K$:

$$f\downarrow_K^G(k) = f(k), \quad k \in K$$

$$f_1\uparrow_K^G(g) = \begin{cases} f_1(g) & g \in K \\ \mathbf{0} & g \notin K \end{cases}$$

where $f\downarrow_K^G \in \mathcal{I}_K$ and $f_1\uparrow_K^G \in \mathcal{I}_G$.

Recall that $s : G/K \to G$ is a function choosing a coset representative $\bar{p}$ for each coset $pK$. Using the restrict operation, the subsampling operation $S_b\downarrow^G_K(f; \Phi)$ in Equation (4) can equivalently be described as:

$$pK = \Phi(f)$$
$$f_b = [\pi(\bar{p}^{-1})f]\downarrow^G_K$$
$$[f_b, \, pK] = S_b\downarrow^G_K(f; \Phi)$$

And the upsampling operation $S_u\uparrow^G_K$ can be rewritten using the extend operation as:

$$f_u = S_u\uparrow^G_K(f_1, pK) = \pi(\bar{p})(f_1\uparrow^G_K)$$

For any $u \in G$ let $f' = \pi(u)f$ and $[f'_b, \, p'K] = \pi'(u)[f_b, \, pK]$ where $\pi$ and $\pi'$ are specified in Equation (3) and Equation (6) respectively. Since $\Phi$ is equivariant, we have

$$\Phi(f') = \Phi(\pi(u)f) = u \cdot \Phi(f) = u \cdot pK = p'K \tag{17}$$

Recall that $\bar{p}'^{-1}u\bar{p} = k_{up}k_p^{-1}$ from Equation (16). Hence $\bar{p}'^{-1} = k_{up}k_p^{-1}\bar{p}^{-1}u^{-1}$ and we have

$$
\begin{aligned}
[\pi(\bar{p}'^{-1})f']\downarrow^G_K &= [\pi(k_{up}k_p^{-1}\bar{p}^{-1}u^{-1})f']\downarrow^G_K \\
&= \pi(k_{up}k_p^{-1})[\pi(\bar{p}^{-1})\pi(u^{-1})f']\downarrow^G_K \\
&= \pi(k_{up}k_p^{-1})[\pi(\bar{p}^{-1})f]\downarrow^G_K \\
&= \pi(k_{up}k_p^{-1})f_b = f'_b
\end{aligned}
\tag{18}
$$

From Equations (17) and (18), $S_b\downarrow^G_K$ is equivariant, i,e.

$$\pi'(u)S_b\downarrow^G_K(f; \Phi) = S_b\downarrow^G_K(\pi(u)f; \Phi)$$

For the upsampling operation, let $[f'_1, p'_1K] = \pi'(u)[f_1, p_1K]$ and $f'_u = \pi(u)f_u$. From Equation (6) we have $f'_1 = \pi(\bar{p}'^{-1}u\bar{p})f_1$. Hence

$$
\begin{aligned}
S_u\uparrow^G_K(f'_1, p'K) &= \pi(\bar{p}')f'_1\uparrow^G_H \\
&= \pi(\bar{p}')[\pi(\bar{p}'^{-1}u\bar{p})f_1]\uparrow^G_H \\
&= \pi(\bar{p}')\pi(\bar{p}'^{-1}u\bar{p})(f_1\uparrow^G_H) \\
&= \pi(u\bar{p})f_1\uparrow^G_H = \pi(u)f_u = f'_u
\end{aligned}
$$

Therefore, $S_u\uparrow^G_K$ is equivariant, i.e.

$$\pi(u)S_u\uparrow^G_K([f_1, p_1K]) = S_u\uparrow^G_K(\pi'(u)[f_1, p_1K])$$

$\square$

### D.3 Proof of Proposition 2.3 (See page 5)

**Proposition 2.3.** *If $S_b\downarrow^G_K : \mathcal{I}_G \to \mathcal{I}_K \times G/K$ (as defined in Equation (4)) is an equivariant map, then the corresponding $\Phi : \mathcal{I}_G \to G/K$ must be equivariant.*

*Proof.* For any $u \in G$, let

$$f' = \pi(u)f$$
$$[f_b, pK] = S_b\downarrow^G_K(f; \Phi)$$
$$[f'_b, p'K] = S_b\downarrow^G_K(f'; \Phi)$$

If $S_b\downarrow^G_K$ is an equivariant map, we have

$$[f'_b, p'K] = \pi'(u)[f_b, pK]$$

Since $\pi'(u)$ corresponds to an invertible transformation, $f_b$, $f'_b$ must contain function values of $f$ at the same set of inputs. Because $\pi(u)f(\cdot) = f(u^{-1}\cdot)$, $f_b$ contains the evaluation of $f$ at $pK$ and $f'_b$ contains the evaluation of $f$ at $u^{-1}p'K$. Therefore, we have

$$p'K = u(pK),$$

i.e. $\Phi$ is equivariant.

Note that the proof above does not require the specific definition of the group action $\pi'$ on the space $\mathcal{I}_K \times G/K$. $\qquad\square$

### D.4 Proof of Proposition B.1 (See page 18)

**Proposition B.1.** $\nu^{-1}$ *is the inverse of* $\nu$*, hence* $\nu$ *is bijective. And* $\forall u \in G$ *we have:*

$$u \cdot \nu(p_1 G_1, p_2 G_2, \ldots, p_L G_L) = \nu(u \cdot (p_1 G_1, p_2 G_2, \ldots, p_L G_L)).$$

*Proof.* Firstly, we prove that $\nu^{-1} \circ \nu$ is an identity map, i.e. $\nu^{-1} \circ \nu = \mathbb{1}_z$. Let $r_{\text{eq}} = \nu(p_1 G_1, p_2 G_2, \ldots, p_L G_L) = (\bar{p}_1 \bar{p}_2 \ldots \bar{p}_L) G_L$ and $(p'_1 G_1, p'_2 G_2, \ldots, p'_L G_L) = \nu^{-1}(r_{\text{eq}})$. From Equation (14), we know that $p'_1 G_L = r_{\text{eq}}$. Hence we can let $p'_1 = \bar{p}_1 \bar{p}_2 \ldots \bar{p}_L g_L$ where $g_L \in G_L$. Since $(\bar{p}_2 \bar{p}_3 \ldots \bar{p}_L) \in G_1$, for $l = 1$ we have

$$\bar{p}'_1 = \bar{p}_1$$
$$p'_2 = \bar{p}'^{-1}_1 \cdot p'_1 = \bar{p}_2 \bar{p}_3 \ldots \bar{p}_L g_L$$

And recursively, for $l = 1, \ldots, L$ we would have

$$\bar{p}'_l = \bar{p}_l$$
$$p'_{l+1} = \bar{p}_{l+1} \ldots \bar{p}_L g_L$$

Hence $(p_1 G_1, p_2 G_2, \ldots, p_L G_L) = (p'_1 G_1, p'_2 G_2, \ldots, p'_L G_L)$ and $\nu^{-1} \circ \nu = \mathbb{1}_z$.

Secondly, we prove that $\nu \circ \nu^{-1}$ is also an identity map, i.e. $\nu \circ \nu^{-1} = \mathbb{1}_r$. Let $(p'_1 G_1, p'_2 G_2, \ldots, p'_L G_L) = \nu^{-1}(r_{\text{eq}})$ and $r'_{\text{eq}} = \nu(p'_1 G_1, p'_2 G_2, \ldots, p'_L G_L)$. From Equation (14), we have $r_{\text{eq}} = p'_1 G_L$ and $p'_l = \bar{p}'^{-1}_{l-1} \cdot p_{l-1}$. Hence

$$r_{\text{eq}} = p'_1 G_L = \bar{p}'_1 p'_2 G_L = \cdots = \bar{p}'_1 \bar{p}'_2 \ldots \bar{p}'_{L-1} p'_L G_L = \bar{p}'_1 \bar{p}'_2 \ldots \bar{p}'_L G_L = r'_{\text{eq}}$$

Therefore $\nu \circ \nu^{-1} = \mathbb{1}_r$ and $\nu$ is bijective.

Lastly, we prove $\nu$ is equivariant. Let $(p'_1 G_1, p'_2 G_2, \ldots, p'_L G_L) = u \cdot (p_1 G_1, p_2 G_2, \ldots, p_L G_L)$ where the group action is implied by Equation (6). From Equation (16), we know that when $u \in G$, $\pi(\bar{p}'^{-1}_1 u \bar{p}_1) \in G_1$. Recursively, we have

$$\bar{p}'^{-1}_l \ldots \bar{p}'^{-1}_2 \bar{p}'^{-1}_1 u \bar{p}_1 \bar{p}_2 \ldots \bar{p}_l \in G_l \tag{19}$$

for $l = 1, \ldots, L$. When $l = L$, from $\bar{p}'^{-1}_L \ldots \bar{p}'^{-1}_2 \bar{p}'^{-1}_1 u \bar{p}_1 \bar{p}_2 \ldots \bar{p}_l \in G_L$, we have

$$\bar{p}'_1 \bar{p}'_2 \ldots \bar{p}'_L G_L = u \cdot (\bar{p}_1 \bar{p}_2 \ldots \bar{p}_L G_L)$$

Hence $u \cdot \nu(p_1 G_1, p_2 G_2, \ldots, p_L G_L) = \nu(u \cdot (p_1 G_1, p_2 G_2, \ldots, p_L G_L))$. So that the group action given by Equation (6) is simplified to the left-action on the single group element. $\qquad\square$

### D.5 Proof of Proposition 3.1 (See page 6)

**Proposition 3.1.** *When* ENC *and* DEC *are given by Equations (8) and (9), and the group actions are specified as in Equation (3) and Equation (6), for any* $g \in G$ *and* $f \in \mathcal{I}_G$*, we have*

$$[z_{inv}, g \cdot z_{eq}] = \text{ENC}(\pi(g)f)$$
$$\pi(g)\hat{f} = \text{DEC}(z_{inv}, g \cdot z_{eq})$$

*Proof.* Let $f, f' \in \mathcal{I}_G$ be the input feature maps where $f' = \pi(g)f$. Let $[f_l, \; p_l G_l]$ and $[f'_l, \; p'_l G_l]$ be the feature maps and subsampling indexes output by the $l$-th subsampling layer for $f$ and $f'$ respectively. Let $[z_{\text{inv}}, z_{\text{eq}}] = \text{ENC}(f)$ and $[z'_{\text{inv}}, z'_{\text{eq}}] = \text{ENC}(f')$ and let $r_{\text{eq}} = \nu(z_{\text{eq}})$, $r'_{\text{eq}} = \nu(z'_{\text{eq}})$ where $\nu$ is given in Equation (13).

From Equation (6) and the equivariance of $\text{G-CNN}_l(\cdot)$, we have
$$f'_1 = \pi(\bar{p}_1'^{-1} g \bar{p}_1) f_1$$
and recursively:
$$f'_l = \pi((\bar{p}'_1 \bar{p}'_2 \ldots \bar{p}'_l)^{-1} g(\bar{p}_1 \bar{p}_2 \ldots \bar{p}_l)) f_l \tag{20}$$
where $l = 1, \ldots, L$ and $(\bar{p}'_1 \bar{p}'_2 \ldots \bar{p}'_l)^{-1} g(\bar{p}_1 \bar{p}_2 \ldots \bar{p}_l) \in G_l$ (see Equation (19)).

Since $G_L = \{e\}$ when $l = L$, we have
$$(\bar{p}'_1 \bar{p}'_2 \ldots \bar{p}'_L)^{-1} g(\bar{p}_1 \bar{p}_2 \ldots \bar{p}_L) = e$$
Hence
$$f'(e) = f(e)$$
$$(\bar{p}'_1 \bar{p}'_2 \ldots \bar{p}'_L) = g \cdot (\bar{p}_1 \bar{p}_2 \ldots \bar{p}_L)$$
which can be rewritten as
$$z'_{\text{inv}} = z_{\text{inv}}$$
$$r'_{\text{eq}} = g \cdot r_{\text{eq}}$$
From Proposition B.1 we have
$$z'_{\text{eq}} = \nu^{-1}(r'_{\text{eq}}) = \nu^{-1}(u r_{\text{eq}}) = g \cdot \nu^{-1}(r_{\text{eq}}) = g \cdot z_{\text{eq}}$$
Therefore, for the encoders we have $[z_{\text{inv}}, g \cdot z_{\text{eq}}] = \text{ENC}(\pi(g)f)$.

For the decoders, let
$$z'_{\text{eq}} = (p'_1 G_1, p'_2 G_2, \ldots, p'_L G_L) = g \cdot z_{\text{eq}}$$

Since the feature map at the $l$-th subsampling layer is transformed according to Equation (20), the sampling index is transformed accordingly:
$$p'_l G_l = (\bar{p}'^{-1}_{l-1} \ldots \bar{p}'^{-1}_2 \bar{p}'^{-1}_1 g \bar{p}_1 \bar{p}_2 \ldots \bar{p}_{l-1}) p_l G_L$$
fFrom the definition of equivariant upsampling in Equation (5), we have
$$f'_{l-1} = \pi(\bar{p}'^{-1}_{l-1} \ldots \bar{p}'^{-1}_2 \bar{p}'^{-1}_1 g \bar{p}_1 \bar{p}_2 \ldots \bar{p}_{l-1}) f_{l-1}$$
where $l = L, \ldots, 1$. When $l = 1$, we have $f'_0 = \pi(g)f_0$ so that $\pi(g)\hat{f} = \text{DEC}(z_{\text{inv}}, g \cdot z_{\text{eq}})$.

$\square$

# E   Additional Experiments

## E.1   Rot-Translation-MNIST Classification

We train a P4CNN model (Cohen and Welling, 2016) on the rotated MNIST dataset (Larochelle et al., 2007). To better explore the effects of subsampling methods, we pad the $28 \times 28$ images to $32 \times 32$, and insert 5 subsampling (max-pooling) layers with a scale factor of 2 between convolutional layers and perform max-pooling on the 4 rotations in the end. This model achieves classification accuracy that is close to the results reported in Cohen and Welling (2016).

We replace these max-pooling layers with our equivariant subsampling layers. The final predictions indeed have perfect invariance to translations and rotations. However, so far we did not observe performance gains, likely because the rotated MNIST dataset is center-aligned, so the exact translation equivariance from our subsampling module offers no advantage over baselines.

To investigate whether our approach offers performance gain when data is not well-aligned, we pad the images to $48 \times 48$ and augment them with translations. We observed that the baseline P4CNN achieves $93.82 \pm 0.08\%$ by training on 12000 examples and test on 50000 held-out examples (using the same data split setting as Cohen and Welling (2016)), while P4CNN with equivariant subsampling achieves $96.43 \pm 0.16\%$, showing that the exact equivariance (and hence invariance for the classifier) leads to noticeable gains in generalisation performance.

Table 3: Stability of sampling indices

| Type of $\Phi$ | Plain Argmax | | | | Argmax after smoothing equivariant layers | | | |
|---|---|---|---|---|---|---|---|---|
| Gaussian noise $\sigma$ | $\sigma = 0.01$ | $\sigma = 0.1$ | $\sigma = 0.2$ | $\sigma = 0.5$ | $\sigma = 0.01$ | $\sigma = 0.1$ | $\sigma = 0.2$ | $\sigma = 0.5$ |
| $x$ | 3.86 | 3.99 | 4.41 | 9.95 | 0.03 | 0.36 | 0.33 | 1.13 |
| $y$ | 3.92 | 4.05 | 4.48 | 9.92 | 0.04 | 0.36 | 0.34 | 1.16 |
| $\cos(x)$ | 0.69 | 0.69 | 0.69 | 0.70 | 0.13 | 0.53 | 0.63 | 0.70 |
| $\sin(x)$ | 0.68 | 0.69 | 0.68 | 0.70 | 0.11 | 0.56 | 0.65 | 0.69 |

### E.2 Stability of sampling indices

We provide an empirical analysis of the stability of $z_{\text{eq}}$ with different choices of $\Phi$ below: We first convert the tuple $z_{\text{eq}}$ to a single group element $r_{\text{eq}}$ (see Appendix B.2), and estimate the standard deviation (std) of $r_{\text{eq}}$ as we add i.i.d. Gaussian noise to the inputs. In the case of $p4$ group, $r_{\text{eq}}$ can be written as a vector $r_{\text{eq}} = [x, y, r]$ where $x, y$ correspond to horizontal/vertical translations and $r$ corresponds to rotations. For the rotation dimension $r$ in $r_{\text{eq}}$, it may not make sense to directly estimate its std, so we instead estimate the std of $\cos(r)$ and $\sin(r)$. The results are shown in Table 3.

The original input feature maps take values within $[0, 1]$, and we add Gaussian noise from $\mathcal{N}(0, \sigma)$ to the inputs. The std $\sigma$ controlling the noise level is specified in the table. The output $x, y$ are in the range of $[0, 63]$, and $r \in \{0, \frac{\pi}{2}, \pi, \frac{3\pi}{2}\}$. As shown in the table, argmax with smoothing equivariant layers (used in our experiments) is relatively stable. The stds of the translation dimension are below one pixel until at least a noise level of $\sigma = 0.2$. The rotation dimension is relatively unstable for all versions of $\Phi$, and we will explore preprocessing equivariant layers that could improve its stability in the future. We also notice that the plain argmax function is indeed unstable, which justify the necessity of inserting equivariant layers before taking the argmax.

Moreover, If we could find a way to differentiate through the sampling index, we could use trainable equivariant layers (possibly parameterised by G-convolutional layers) in $\Phi$. However, how to design such differentiable sampling indices is not trivial, and we consider exploring this in the future.

## F Implementation Details

In this section, we outline a few important implementation details, and leave other details to the reference code at https://github.com/jinxu06/gsubsampling.

### F.1 Data

For Colored-dSprites and FashionMNIST datasets, we add colours to grayscale images from Matthey et al. (2017) and Xiao et al. (2017) by sampling random scaling for each channel uniformly between $0.5$ and $1$ following Locatello et al. (2019). For FashionMNIST, we also apply zero-paddings to images to reach the size of $64 \times 64$. We then translate the images with random displacements uniformly sampled from $\{(x, y) | -18 \le x, y \le 18, x, y \in \mathbb{Z}\}$, and rotate the images with uniformly sampled degrees from $\{\frac{360 \times k}{32} | k = 0, \ldots, 32\}$. We use the original Multi-dSprites dataset as provided in Kabra et al. (2019). For CLEVR6, we crop images from the original CLEVR (Johnson et al., 2017) at y-coordinates $(29, 221)$ bottom and top, and at x-coordinates $(64, 256)$ left and right as stated in Burgess et al. (2019). We then resize the images to $64 \times 64$ so that we can use the same model for both multi-object datasets. We only use images with up to 6 objects in CLEVR following Greff et al. (2019). For evaluation, we use a randomly sampled test set with 10000 examples that has no overlap with training data for all datasets. In Figure 6, we show examples from different datasets.

### F.2 Model Architectures

**The equivariant map $\Phi$** One can insert arbitrary equivariant layers before and after we take the norm $\| \cdot \|_1$ in Equation (7). In experiments, we apply mean subtraction followed by average pooling with kernel size 5 before taking the norm, and apply Gaussian blur with kernel size 15 after taking the norm. They are inserted in the purpose of smoothing feature maps and avoiding non-unique argmax when possible (in Appendix B.1 we discuss the case when non-unique argmax cannot be avoided). In

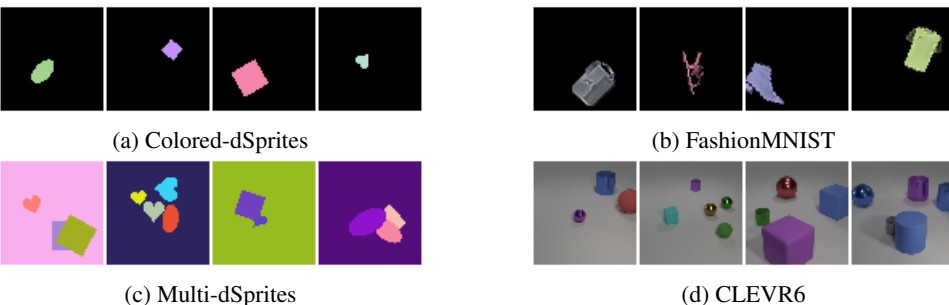

(a) Colored-dSprites                    (b) FashionMNIST

(c) Multi-dSprites                      (d) CLEVR6

Figure 6

practice, we use Equation (15) to obtain all subsampling indexes at the same time rather than passing through $\Phi$ multiple times.

**Autoencoders**  For all single object experiments, we use 5 layers of ($G$-)equivariant convolutional layers in encoders, and the decoders mirror the architecture of the encoders except for the output layers. In baseline models, we use strided convolution as a way to perform subsampling/upsampling, while in equivariant models we use equivariant downsampling/upsampling. We rescale the number of channels such that the total number of parameters of the models roughly match one another. However, exact correspondence is not achievable because exact equivariant models use equivariant subsampling to transform feature maps into vectors at the final layer of the encoder, while baseline models apply flattening. Please see the reference implementation for other details about network architectures.

We use scale factor 2 for all subsampling and upsampling layers in baseline models. For GAE-$p1$, the feature maps at each layer are defined on the following chain of nested subgroups: $\mathbb{Z}^2 \geq (2\mathbb{Z})^2 \geq (4\mathbb{Z})^2 \geq (8\mathbb{Z})^2 \geq (16\mathbb{Z})^2 \geq \{e\}$. For For GAE-$p4$, we use $\mathbb{Z}^2 \rtimes \mathsf{C}_4 \geq (2\mathbb{Z})^2 \rtimes \mathsf{C}_4 \geq (4\mathbb{Z})^2 \rtimes \mathsf{C}_4 \geq (8\mathbb{Z})^2 \rtimes \mathsf{C}_4 \geq (16\mathbb{Z})^2 \rtimes \mathsf{C}_2 \geq \{e\}$. And for GAE-$p4m$, we use $\mathbb{Z}^2 \rtimes (\mathsf{C}_4 \rtimes \mathsf{C}_2) \geq (2\mathbb{Z})^2 \rtimes (\mathsf{C}_4 \rtimes \mathsf{C}_2) \geq (4\mathbb{Z})^2 \rtimes (\mathsf{C}_4 \rtimes \mathsf{C}_2) \geq (8\mathbb{Z})^2 \rtimes (\mathsf{C}_4 \rtimes \mathsf{C}_2) \geq (16\mathbb{Z})^2 \rtimes (\mathsf{C}_2 \rtimes \mathsf{C}_2) \geq \{e\}$.

**Object discovery**  For baseline models, we adopt the exact same architecture as the original MONet (Burgess et al., 2019) using the implementation provided by Engelcke et al. (2020). For MONet-GAEs, we simply replace Component VAEs in the original MONet with our V-GAEs. Both Component VAEs and V-GAEs have a latent size of 16.

### F.3  Hyperparameters

For all single object experiments, we use Adam optimizer (Kingma and Ba, 2015) with a learning rate of 0.0001 and a batch size of 16. We use 16-bits precision to enable faster training and reduce memory consumption. For experiments on multi-object datasets, hyperparameters will match the original MONet (Burgess et al., 2019) except that we still use a batch size of 16 instead of 64 stated in the original paper. This is because we observed that in the low data regime, batch size 16 trains faster and performs no worse than batch size 64 for the problems we considered here.

### F.4  Computational Resources

In theory, the only computational overhead is caused by computing sampling indices, which is negligible compared to the forward pass of ($G$-)Convolutional layers. In practice, our current implementation uses `torch.gather` to perform subsampling, and relies on for-loops over data batches when applying group actions to feature maps, which we believe can be made more efficient. Hence on a single GeForce GTX 1080 GPU card, a standard GAE-p1 takes around 30 minutes to train for $100k$ steps, compared to 16 minutes for standard ConvAEs.

## G  Societal Impacts

This work presents group equivariant subsampling and upsampling operations, which can be combined with lifting-based group equivariant neural networks to construct more computational efficient

equivariant models. Therefore, it can potentially cut down energy consumption and carbon footprint for training these models. Furthermore, because these operations are exactly equivariant, they can improve sample complexity in many scenarios, which further reduce model training time and require less data collection. However, as we have shown in Section 5, group equivariant autoencoders constructed using these subsampling/upsampling layers enable us to manipulate reconstructions. Hence they can be used to generate fake images that have many potential malicious usages.