# OpenReview forum: "Group Equivariant Subsampling"
_NeurIPS.cc/2021/Conference — NeurIPS 2021 Poster_

### Official Review · Reviewer_hKzy · 2021-07-14

**Rating:** 8
**Confidence:** 5

**Summary:**

The paper proposes and evaluates equivariant subsampling and upsampling operations for group convolutional networks, which include vanilla CNNs as a special case. These operations address the issue that conventional subsampling and upsampling operations are only equivariant under the action of a subgroup of symmetries. The models have the property to disentangle the appearance of objects from their G-pose, which allows to build disentangled group equivariant autoencoders.

As a simple introduction, the authors formulate their method initially for the special case of $G = \mathbb{Z}$ being the 1-dimensional translation group. In contrast to conventional subsampling operations, which sample the values on a grid $c\mathbb{Z}$ with spacing $c$, the equivariant subsampling operation samples features on a shifted grid $c\mathbb{Z} + i$, where $i = 0, \dots, c-1$ is an integer offset that is predicted from the signal itself via some equivariant function. The tuple of subsampled features and the integer offset are shown to transform in an equivariant way under the action of the full symmetry group $\mathbb{Z}$. If multiple subsampling layers are successively applied in a deep network, each of them adds a new offset value to a sequence of offsets. When eventually pooling to the trivial group $G=\{e\}$, the resulting feature is position invariant and all position information is encoded in the offset sequence. Upsampling layers are accordingly taking a low-resolution signal and an offset and embed it with this offset into a higher-resolution grid.

These constructions are in the obvious way generalized to feature maps on arbitrary discrete groups $G$: The lower resolution sampling grid is here given by a coset $pK \in G/K$ of a subgroup $K<G$ in $G$. The group equivariant subsampling operation is therefore producing a tuple which consists of a subsampled signal on the coset and the coset (offset) itself. To upsample such a signal, it is embedded in the coset of the supergroup. Lemma 2.1 and Proposition 2.2 assert the full $G$-equivariance of these operations.

The integer offset, or, more generally, coset, needs to be computed via any equivariant map from the input signal to the space of cosets. A simple choice, made by the authors, is that coset $gK$ that consist to the argmax group element $g$ of the signal's l1-norm.

The equivariant subsampling layers are used to construct various group equivariant autoencoders (GEAs), which are evaluated against baseline autoencoders that are constructed from group convolutions but use conventional subsampling and upsampling layers. GEAs are empirically shown to be exactly equivariant and to generalize perfectly out of distribution. This is not the case for the baseline models, whose conventional subsampling and upsampling layers break the models' overall equivariance. The disentanglement of object appearance and $G$-pose in the autoencoders' latent space is in figure 3 (left) shown to allow for an independent manipulation of these factors. GEAs perform on single and multi-object reconstruction tasks significantly better than the baselines.

**Limitations And Societal Impact:**

The authors discuss limitations of the method appropriately. No immediate negative societal impacts are to be expected.

**Main Review:**

The submission addresses an important open problem of conventional and group equivariant convolutional networks, namely the lack of equivariance in their downsampling and upsampling layers. While other common types of layers like convolutions, nonlinearities or batch normalization have equivariant counterparts, contemporary models are usually using non-equivariant or smoothing based subsampling layers, which prevents the exact equivariance of the full model. The proposed method solves this issue, allowing to build fully equivariant models. As demonstrated, this can be used to disentangle an object's pose from its appearance and generally improves the model performance.

The paper is well written and mathematically sound. All claims are proven in the appendix and demonstrated empirically. The decision to introduce the construction on the simple and practically relevant special case of $G=\mathbb{Z}$ being the translation group helps to make the idea accessible to a broader audience. I am confident to be able to reimplement the model and reproduce the experiments.

While any equivariant map is suitable to predict the grid offset (or coset), the authors choose the argmax of the feature vectors' norms. This choice seems to be very unstable under noise on the input, which should be investigated empirically. What happens in degenerate cases like reconstructing the top row in figure 3 (right) with $z_\text{eq}$ taken from the bottom row, whose ellipses are $C_2$-symmetric?
A more robust choice of equivariant map could be based on coset average pooling, followed by an argmax over cosets or on the phase offset of a harmonic basis function expansion of the features on $G$. The authors should discuss different choices of such maps and empirically investigate their stability.

It seems intuitively necessary that equivariant sub/upsampling operations are based on an offset that is computed from the signal itself - can this conjecture be proven formally?

While the experimental section is convincingly demonstrating the merits of the method, it does not include a standard task like image classification. It would be interesting to see the performance gains on such tasks instead of only auto-encoding reconstruction errors.

Overall, the submission solves an important problem of equivariant networks, and I would be happy to give it a very positive score. My low score should be understood as an incentive to extend the empirical evaluation, in particular on the choice of the equivariant offset predictor.

**Time Spent Reviewing:**

6

---

> ### Author Response · Authors · 2021-08-11
> **Response to Reviewer hKzy**
>
> We are grateful to receive your thoughtful feedback with constructive criticism. They are very helpful for improving this work. Below are our responses to your suggestions and questions:
>
> *“This choice seems to be very unstable under noise on the input, which should be investigated empirically … The authors should discuss different choices of such maps and empirically investigate their stability”*
>
> Because our subsampling methods only require $\Phi$ (map from feature space to the quotient $G/K$) to be equivariant, one can insert arbitrary equivariant layers to smooth the input features before taking the argmax of l1-norm in eq.11. In our experiments, we inserted average pooling and Gaussian blur (as described in E.1), and they work well enough in practice. These operations will alleviate the instability of these computed sampling indexes (cosets).
>
> We thus provide an empirical analysis of stability of $z_{eq}$ with different choices of $\Phi$ below: We first convert the tuple $z_{eq}$ to a single group element $r_{eq}$ (see Line 197-200 and B.2), and estimate the standard deviation (std) of $r_{eq} = [x, y, r]$ (in the case of $p4$ group) as we add i.i.d. Gaussian noise to the inputs. For the rotation dimension $r$ in $r_{eq}$, it may not make sense to directly estimate its std, so we instead estimate the std of $\cos(r)$ and $\sin(r)$. Below are the results:
>
> | Gaussian noise std $\sigma$ | $\sigma=$0.01 | $\sigma=$0.1  | $\sigma=$0.2 | $\sigma=$0.5 |
> | ------------- |:-------------:| :----------:| : ----------:|  : ----------:|
> | std for plain argmax  | 3.86 (x), 3.92 (y), 0.69 (cos(r)), 0.68 (sin(r)) | 3.99, 4.05, 0.69, 0.69 | 4.41, 4.48, 0.69, 0.68 | 9.95, 9.92, 0.70, 0.70 |
> | std for argmax after smoothing equivariant layers (adopted in the paper)  | 0.03, 0.04, 0.13, 0.11 | 0.36, 0.36, 0.53, 0.56 | 0.33, 0.34, 0.63, 0.65 | 1.13, 1.16, 0.70, 0.69 |
> | std for argmax after coset average pooling (as suggested) | 6.09, 6.05, 0.52, 0.52 | 14.88, 15.01, 0.70, 0.71 | 16.70, 16.75, 0.71, 0.71 | 18.00, 18.10, 0.71, 0.71 |
>
> The four numbers in each cell are the estimated std of $x, y, \cos(r), \sin(r)$ correspondingly. The original input values are within $[0, 1]$, and the noise is drawn from $\mathcal{N}(0,\sigma)$ with $\sigma$ specified in the table controlling the noise level. The output $x, y$ are in the range of [0, 63], and $r = 0,\frac{\pi}{2}, \pi \text{ or } \frac{3\pi}{2}$. We use argmax with smoothing equivariant layers in practice, which is relatively stable as shown in the table. The stds of the translation dimension are below one pixel until at least a noise level of $\sigma=0.2$. The rotation dimension is relatively unstable for all versions of $\Phi$, and we will explore preprocessing equivariant layers that could improve its stability in the future. We also notice that the plain argmax function is indeed unstable, which justify the necessity of inserting equivariant layers before taking the argmax. The coset average pooling approach as suggested is relatively unstable, and we think this is because features pooled over each coset are quite close due to the averaging, so noise can easily influence which coset has the largest feature norm.
>
> Furthermore, if we could find a way to differentiate through the sampling index, we could use trainable equivariant layers (possibly parameterised by G-convolutional layers) in $\Phi$. However, how to design such differentiable sampling indexes is not trivial, and we consider exploring this in the future.
>
> Overall, we agree with the reviewer that this is an important problem that should be further investigated, there may exist better solutions.
>
> *“What happens in degenerate cases like reconstructing the top row in figure 3 (right) with
>  taken from the bottom row, whose ellipses are symmetric?”*
>
> For degenerate cases where inputs are symmetric, the argmax returns a set of sampling indices rather than a single one (as discussed in B.1). If we instead consider including this set of sampling indices (rather than a single one) in $z_{eq}$, and let group acts on this set, it can be shown that the exact equivariance would still hold. When performing subsampling, We can use any sampling index from this set to perform subsampling, and the output will always be the same. This complexity is unavoidable because the equivariant map that maps the feature map to a single coset does not exist in this case.
>
> *“It seems intuitively necessary that equivariant sub/upsampling operations are based on an offset that is computed from the signal itself - can this conjecture be proven formally?”*
>
> In our framework we define subsampling as a restriction of a function on $G$ to a coset $pK$, and we've shown that if the $\Phi$ that maps $f$ to $pK$ is equivariant, then subsampling is equivariant. We can in fact prove the converse, that if subsampling is equivariant then $\Phi$ must be equivariant. A sketch of the proof is as follows:
>
> Let $f$ and $f’$ be the input feature map before and after transformation $\pi(u)$, and let $[f_b, pK] = S_b(f; \Phi), [f’_b, p’K] = S_b(\pi(u)  f; \Phi)$. Recall that $S_b$ is our equivariant subsampling operation (eq.8), and $f_b,f’_b$ are the subsampled feature maps defined on the subgroup $K$. If there exists a group action $\pi^{\prime}$ s.t. $[f’_b, p’K] = \pi’(u) [f_b, pK] $, then $f_b, f’_b$ must contain function values of the input feature map $f$ evaluated at the same set of inputs, otherwise a one-to-one map cannot be established between $[f_b, pK]$ and $[f’_b, p’K]$ as one would contain information that the other does not have. Because $\pi(u) f(\cdot) = f(u^{-1} \cdot )$, $f_b$ contains the evaluation of $f$ at $pK$ and $f_b$ contains the evaluation of $f$ at $u^{-1} p’K$. Therefore, $p’ K = u (pK)$ and $\Phi$ is equivariant.
>
> The above implies that $\Phi$ must depend on $f$ (since our equivariance is defined by the regular representation action on the spaces $I_G$ and $I_K\times G/K$).
>
> *“It does not include a standard task like image classification.”*
>
> We ran some preliminary experiments on rotated MNIST, and below are our findings so far: We train a P4CNN model on the rotated MNIST dataset. To better explore the effects of subsampling methods, we pad the $28\times 28$ image to $32\times 32$, and insert 5 subsampling (max-pooling) layers with a factor of 2 and max-pool 4 rotations in the end. This model achieves classification accuracy which is close to the results reported in (Cohen et al 2016). We replace these max-pooling layers with our equivariant subsampling ones. The final predictions indeed have perfect invariance to translations and rotations. So far we did not observe performance gains, likely because the MNIST dataset is center-aligned, so the exact translation equivariance from our subsampling module offers no advantage over baselines. To investigate whether our approach offers performance gain when data is not well-aligned, we pad the images to $48\times 48$ and augment them with translations. We observed that the baseline P4CNN achieves $93.82\pm0.08$% by training on 12k examples and test on 50k held-out examples (same setting as in Cohen et al 2016), while P4CNN with equivariant subsampling achieves $96.43\pm0.16$%, showing that the exact equivariance (and hence invariance for the classifier) leads to noticeable gains in generalisation performance. We will include these classification results in the next version of the paper.

---

> > ### Comment · Reviewer_hKzy · 2021-08-17
> > **Raised score**
> >
> > I thank the authors for their detailed response. As it addressed all of my concerns, I raised my score to 8.

---

### Official Review · Reviewer_RUmB · 2021-07-15

**Rating:** 7
**Confidence:** 4

**Summary:**

The authors propose subsampling and upsampling layers that are equivariant to common transformation groups. These constructions are used to design autoencoders consisting of layers that alternate between group equivariant convolutions and group equivariant subsampling/upsampling layers. The result is that the entire autoencoder is *exactly* group equivariant. Experiments on translations, translations+rotations, translations+rotations+reflections for simple images with single/multiple objects clearly show the improved performance over subsampling methods that are not group-equivariant.

**Ethics Review Area:**

["I don’t know"]

**Limitations And Societal Impact:**

The authors have discussed the limitations of the paper.

**Main Review:**

The core of the paper is group equivariant subsampling method that is clever and simple to implement. It can be easily integrated with the architectures which already employ group equivariant convolutions and can be composed with each other to form a deep network. Although the simple method shown in Eqn. 11 may not always have a unique solution, as in the case of feature maps with symmetries, the authors propose a way to overcome this problem.

The main weakness in the paper would be that in all the experiments, the feature maps are all images on the plane. I would have liked to see how easy or difficult it is to integrate these subsampling layers in the case of spherical image inputs with the feature maps defined on SO(3). And whether this would lead to improved performance in practice. It would also be useful to other researchers to add an another line in the tables with coset pooling as the subsampling operation to see how it affects final reconstruction performance.

The practical significance of this method also deserves more discussion in the paper. The experiments are okay in terms of demonstrating the main strengths of the proposed work. However, I would have like at least one experiment with a more realistic dataset, like a dataset of human faces perhaps undergoing some deformations. Related to this, what about larger transformation groups like piece-wise affine transforms or very large groups like 2D diffeomorphisms (used in facial expression modeling and medical image registration)? My guess is constructing the group-equivariant convolutions for these groups is the much bigger problem rather the group-equivariant subsampling itself.

The authors do mention that for objects with background, the first step is to remove the background before group-equivariant autoencoding and that local symmetries become more important as most scenes have several objects. These are indeed challenging tasks and perhaps this paper can provide a starting point to attack these problems.

The related work section should include other works which are similar in spirit to group equivariant autoencoders. This includes the following list of papers that is not exhaustive:

1. Deforming autoencoders: Unsupervised Disentangling of Shape and Appearance ECCV 2018 https://openaccess.thecvf.com/content_ECCV_2018/html/Zhixin_Shu_Deforming_Autoencoders_Unsupervised_ECCV_2018_paper.html

2. Extracting invariant features from images using an equivariant autoencoder, ACML 2018, http://proceedings.mlr.press/v95/kuzminykh18a.html

3. Rotation-Invariant Autoencoders for Signals on Spheres, https://arxiv.org/abs/2012.04474

4. Capsule networks

**Time Spent Reviewing:**

6 hours

---

> ### Author Response · Authors · 2021-08-11
> **Response to Reviewer RUmB**
>
> We appreciate your time spent with our submission, and would like to thank you for your positive feedback with constructive suggestions. Here are our responses:
>
> *“I would have liked to see how easy or difficult it is to integrate these subsampling layers in the case of spherical image inputs with the feature maps defined on SO(3).”*
>
> It is one of our main future directions to apply this framework empirically to more sophisticated groups such as Lie groups, in particular  3D groups such as SO(3), as mentioned in the Limitations and Future Work section. While the equivariance properties of subsampling/upsampling layers still hold for these cases, we still need to tackle the practical complexities of these applications, e.g. equivariance error that arises due to sampling the signal on G at finitely many group elements (https://arxiv.org/abs/2002.12880, https://arxiv.org/abs/1909.12057). However, we agree that experiments on SO(3) would certainly improve our paper, and we will work towards getting such results.
>
> *“It would also be useful to other researchers to add an another line in the tables with coset pooling as the subsampling operation to see how it affects final reconstruction performance.”*
>
> Indeed there is nothing preventing us from comparing against coset pooling, but for translation equivariance we expect performance to drop significantly with coset pooling because it amounts to pooling over a checkerboard pattern (c.f. Figure 2), which breaks the locality of subsampling. Hence in previous work, coset pooling is typically only used for rotations, while conventional subsampling is used for translations, making the approach only approximately equivariant.
>
> *“I would have like at least one experiment with a more realistic dataset, like a dataset of human faces perhaps undergoing some deformations. Related to this, what about larger transformation groups like piecewise affine transforms or very large groups like 2D diffeomorphisms (used in facial expression modeling and medical image registration)?”*
>
> It is our important future work to extend this approach to continuous groups, and this would allow the treatment of larger groups such as affine transformations as mentioned. However, because of the practical complexities that arise with these applications, e.g. equivariance error that arises due to sampling the signal on G at finitely many group elements (https://arxiv.org/abs/2002.12880, https://arxiv.org/abs/1909.12057), this may be beyond the scope of what we can realistically achieve in the discussion period.
>
> Although only slightly more realistic than d-Sprites, we have conducted some classification experiments on rotated MNIST for the rebuttal. We train a P4CNN model on the rotated MNIST dataset. To better explore the effects of subsampling methods, we pad the $28\times 28$ image to $32\times 32$, and insert 5 subsampling (max-pooling) layers with a factor of 2 and max-pool 4 rotations in the end. This model achieves classification accuracy which is close to the results reported in (Cohen et al 2016). We replace these max-pooling layers with our equivariant subsampling ones. The final predictions indeed have perfect invariance to translations and rotations. So far we did not observe performance gains, likely because the MNIST dataset is center-aligned, so the exact translation equivariance from our subsampling module offers no advantage over baselines. To investigate whether our approach offers performance gain when data is not well-aligned, we pad the images to $48\times 48$ and augment them with translations. We observed that the baseline P4CNN achieves $93.82\pm0.08$% accuracy by training on 12000 examples and test on $50000$ held-out examples (same setting as in Cohen et al 2016), while P4CNN with equivariant subsampling achieves $96.43\pm0.16$% accuracy, showing that the exact equivariance (and hence invariance for the classifier) leads to noticeable gains in generalisation performance. We will include these classification results in the next version of the paper.
>
> *“The authors do mention that for objects with background ... These are indeed challenging tasks and perhaps this paper can provide a starting point to attack these problems.”*
>
> We agree that dealing with multiple objects and backgrounds is indeed an important future work that could bring the proposed approach to more real-world applications. As a starting point, we would like to outline the following two directions:
>
> (a) As discussed in Line 197-200, $z_{eq}$ can be equivalently expressed as a single group element. A natural extension is that we could consider multiple dimensions $z_{eq}$, such that each dimension is a group element associated with one object. Our equivariance is global because $z_{eq}$ depends on all input dimensions. If we could use an equivariant map $\Phi$ that only has local dependency (probably by masking out pixels out of a certain range), we could have a preliminary form of local equivariance.
>
> (b) Rather than tackling object discovery and background removal within the proposed GAE model, it is also possible to use our model as components in a more complicated architecture. For example, in our paper, we have combined our methods with existing object-centric learning models, such as MONet, so that GAEs only serve as a component modelling each object discovered by other modules.
>
> *“The related work section should include other works which are similar in spirit to group equivariant autoencoders”*
>
> We thank the reviewer for the suggested references, and will include them in our revised version.

---

> > ### Comment · Reviewer_RUmB · 2021-08-25
> > **Good author response**
> >
> > The authors have addressed all the questions that I had. It would be good to add some of this discussion to the main paper so we can better understand future directions, challenges and limitations. Hopefully these ideas can be translated to more realistic settings. I am keeping my score as is.

---

### Official Review · Reviewer_WgFm · 2021-07-16

**Rating:** 6
**Confidence:** 4

**Summary:**

The paper proposes a method to equip discrete-group convolutional neural networks (GCNNs) with a perfectly equivariant subsampling layer. The method relies on a non-fixed sampling grid which is dynamically chosen as an equivariant function of the current input. This method enables encoder-decoder architectures where each layer is equivariant to the global action of the discrete group on the input of the model. Experimental results confirm the accurate equivariance and the improved generalization of the models.

**Limitations And Societal Impact:**

I don't see any issue with the authors' comments on the societal impact of their work.

**Main Review:**

The proposed equivariant subsampling is a clear imrpovement over common down-sampling methods. The main idea seems intuitive and simple to implement, making it probably very easy to adopt in many existing equivariant models.

I found the presentation quite clear overall and I liked the intuitive description of the subsampling idea using the 1D translations as an example. However, I think the introduction to the abstract groups setting was a bit fast; to keep the paper self contained, I think it would be useful to at least define group convolution in the main paper.

There are a number of issues that I think should be addressed more explicitly (see below).

While the reconstruction tasks are certainly important and clearly show the benefit of using this subsampling method, I think some simple experiment on a standard benchmark such as MNIST-rot (for classification) should be included to compare the proposed method with other equivariant models from the literature.

I think the discussion in B.1 should be included more prominently in the main paper. I find the stochastic alternative proposed in B.1 less satisfying. This is not necessarily an issue since I find the subsampling method proposed in the main paper a very nice contribution, but I believe this limitation should be discussed more explicitly in the main paper.

The case of symmetric inputs appears also in Fig. 3 for the ellipse and the square, so some comments about this should also be included in the experimental section.

If I understand correctly, the proposed subsampling method guarantees perfect global equivariance but can reduce the local equivariance. This is because the sampling grid depends on the global image. This means that changing a single pixel in a region of the image far away from a local patter can change the pixels selected around that local patter for the downsampling. Because the downsampling is also performed without any smoothing of the features, this can potentially lead to a completely different prediction for that pattern. Clearly, this is a worst-case scenario and it is not clear yet how this affects the stability of the model in practice.
Still, some comments about this potential instability should be included among the limitations.

Finally, I wonder how this subsampling method affects the performance of a model when using mini-batches, since different samples in the batch might require different sampling grids.

EDIT: Having read the author response, I have decided to maintain my rating.

**Time Spent Reviewing:**

2

---

> ### Author Response · Authors · 2021-08-11
> **Response to Reviewer WgFm**
>
> We would like to thank you for your thoughtful suggestions regarding both writing and experiments. Below are our responses:
>
> *“Some simple experiment on a standard benchmark such as MNIST-rot (for classification) should be included to compare the proposed method with other equivariant models from the literature”*
>
> We ran some preliminary experiments on rotated MNIST, and below are our findings so far: We train a P4CNN model on the rotated MNIST dataset. To better explore the effects of subsampling methods, we pad the $28\times 28$ image to $32\times 32$, and insert 5 subsampling (max-pooling) layers with a factor of 2 and max-pool 4 rotations in the end. This model achieves classification accuracy which is close to the results reported in (Cohen et al 2016). We replace these max-pooling layers with our equivariant subsampling ones, giving the classifier perfect invariance to translations and rotations. With this change only, we did not observe performance gains, likely because the MNIST dataset is center-aligned, so the exact translation equivariance from our subsampling module offers little to no advantage over baselines. To investigate whether our approach offers performance gains when data is not well-aligned, we pad the images to $48\times 48$ and augment them with translations. We observed that the baseline P4CNN achieves $93.82\pm0.08$% by training on 12k examples and test on $50k$ held-out examples (same as Cohen et al 2016), while P4CNN with equivariant subsampling achieves $96.43\pm0.16$%, showing that the exact equivariance (and hence invariance for the classifier) leads to noticeable gains in generalisation performance. We will include these classification results in the next version of the paper.
>
> *“I think the discussion in B.1 should be included more prominently in the main paper. I find the stochastic alternative proposed in B.1 less satisfying.”*
>
> We will add a discussion of symmetric inputs (where the argmax is not unique) in the main paper. We believe that this may not pose a serious issue in practice, because perfectly symmetric inputs are very rare for real-world applications, even though this degenerate case sometimes happens for synthetic data. Furthermore, as discussed in B.1, for symmetric inputs, the equivariant map $\Phi$ would give a set of sampling indices (cosets) rather than a single one. If we instead consider including this set of sampling indices (rather than a single one) in $z_{eq}$, and let group acts on this set, it can be shown that the exact equivariance would still hold. When performing subsampling, We can use any sampling index from this set to perform subsampling, and the output will always be the same. This complexity is unavoidable because an equivariant map that maps the feature map to a single coset does not exist in this case.
>
> *“The case of symmetric inputs appears also in Fig. 3 for the ellipse and the square, so some comments about this should also be included in the experimental section.”*
>
> Thank you for the suggestion. We will also include a discussion of symmetric inputs in the experiment section.
>
> *“This means that changing a single pixel in a region of the image far away from a local patter can change the pixels selected around that local patter for the downsampling … Clearly, this is a worst-case scenario and it is not clear yet how this affects the stability of the model in practice.”*
>
> We acknowledge that pixels far away can affect which local region is selected for downsampling, hence a "smoother" choice of equivariant $\Phi$, that changes smoothly even with such far away perturbations, would be more desirable. One example of such smoother $\Phi$ is that: One can first construct a density function p(g) over $G$ from the input feature maps f(g), where p(g) is also an equivariant feature map. We can then compute the matrix expectation $m=E_{p(g)}[\pi(g)]$ where $\pi(g)$ denotes a matrix group representation of $G$. If this matrix expectation is in the general linear group $GL(n, R)$ (only requires invertibility), then $\pi(G)$ will be a subgroup of $GL(n, R)$, and we can map $m$ to its coset representative in $\pi(G)$. The above construction would give us an equivariant map from the feature space to $G$, and is smoother because it considers the influence of all features in the input feature map. We did not adopt this choice in practice since the simple argmax worked well empirically.
>
> *“I wonder how this subsampling method affects the performance of a model when using mini-batches, since different samples in the batch might require different sampling grids.”*
>
> Indeed it is a good spot that sampling grids are different for data points within one mini-batch, and so we cannot use the same  indexing across examples in a batch to perform subsampling. Instead, we use torch.gather to implement this behaviour. Because of this, a standard GAE-p1 takes around 30 minutes to train for 100k steps on a single GeForce GTX 1080 GPU, compared to 16 minutes for standard ConvAEs, giving a moderate increase in computation time. We will include this discussion in the next version of the paper.

---

> > ### Comment · Reviewer_WgFm · 2021-08-17
> > **Response**
> >
> > Thanks for the clarification and running additional experiments. I am satisfied with these and will maintain my rating as is.

---

### Official Review · Reviewer_a1LF · 2021-07-20

**Rating:** 7
**Confidence:** 4

**Summary:**

This paper introduces a group-theoretic framework for equivariant subsampling. Based on a similar approach to ref [3] in the case of translations, the idea is to subsample according to intrinsic landmarks and keeping track of the shifts involved. The framework introduced can handle this procedure for general symmetry groups by seeing the subsampled space as a subgroup, and the shifts are stored as elements of the quotient of the original group and the subgroup.
The method is evaluated in the context of VAEs, where it is shown that an architecture using equivariant subsampling and upsampling is capable of fully disentangling covariant from invariant features, and, depending on the group used, is shown to generalize perfectly to unseen object positions.

**Limitations And Societal Impact:**

Some limitations were discussed.
I do not foresee potential negative societal impact of this work.

**Main Review:**

Strangely, my main criticism of this paper is also the main reason I find it interesting. A more pedestrian way of describing what has been done is registration of the signal among a collection of possible transformations, according to some intrinsic signal property, while not forgetting the selected registering transformation. In this sense the group-theoretic approach is (an even somewhat restrictive) syntactic sugar. That said, I consider this a neat framework that can represent many previous approaches, that guides implementations, is shown to work in experiments, and may provide leads for extensions in the future. Also, it shows an example of a non-trivial equivariant transformation, where part of the information is absorbed into an index.

It should be noted that registration is inherently unstable no matter what procedure is used to determine the shift (see e.g. https://arxiv.org/abs/1203.1513 2.1ff). There will always be a signal exhibiting the discontinuity of the map from signal to shift parameter (consider a signal with two local maxima in the l1-norm used here, with one peak epsilon/2 lower than the other, but far away in terms of position. A perturbation increasing the lower peak by epsilon will make the registration landmark jump. In general, a map between metric spaces of different dimensionality cannot be continuous in both directions). It may be interesting to explore this a little bit to see whether there are adversarial examples with practical impact.

The groups explored here, while providing a proof of concept, are really not very interesting. I would have been much more delighted to see this framework used to tackle e.g. continuous, or at least more fine-grained discrete rotational steps. The VAE data set samples, by virtue of being one shape on a constant background, would not pose any difficulty (i.e. border effects from e.g. periodic wrap-around) in applying these.

Given the memorization of shifts, can you envision a way of being able to sample from the GAE latent? Would it require putting a probability measure on the shifts?

**Time Spent Reviewing:**

3

---

> ### Author Response · Authors · 2021-08-10
> **Response to Reviewer a1LF**
>
> Thank you for reviewing our submission and provide constructive feedback. Here are our responses to your suggestions and questions:
>
> *“A more pedestrian way of describing what has been done is registration of the signal among a collection of possible transformations, according to some intrinsic signal property, while not forgetting the selected registering transformation. In this sense, the group-theoretic approach is (an even somewhat restrictive) syntactic sugar.”*
>
> We thank the reviewer for pointing us to this body of work that we weren't aware of. We'll include appropriate references for the next version of the paper, and would appreciate it if the reviewer could suggest more citations.
>
> *“It should be noted that registration is inherently unstable no matter what procedure … A perturbation increasing the lower peak by epsilon will make the registration landmark jump.”*
>
> Because our subsampling methods only require $\Phi$ (map from feature space to the quotient $G/K$) to be equivariant, one can insert arbitrary equivariant layers to smooth the input features before taking the argmax of l1-norm in eq.11. In our experiments, we inserted average pooling and Gaussian blur (as described in E.1), and they work well enough in practice. These operations will alleviate the instability of these computed sampling indexes (or registration in the review).
>
> We thus provide an empirical analysis of stability of $z_{eq}$ with different choices of $\Phi$ below: We first convert the tuple $z_{eq}$ to a single group element $r_{eq}$ (see Line 197-200 and B.2), and estimate the standard deviation (std) of $r_{eq} = [x, y, r]$ (in the case of $p4$ group) as we add i.i.d. Gaussian noise to the inputs. For the rotation dimension $r$ in $r_{eq}$, it may not make sense to directly estimate its std, so we instead estimate the std of $\cos(r)$ and $\sin(r)$. Below are the results:
>
> | Gaussian noise std $\sigma$ | $\sigma=$0.01 | $\sigma=$0.1  | $\sigma=$0.2 | $\sigma=$0.5 |
> | ------------- |:-------------:| :----------:| : ----------:|  : ----------:|
> | std for plain argmax  | 3.86 ($x$), 3.92 ($y$), 0.69 ($\cos(r)$), 0.68 ($\sin(r)$) | 3.99, 4.05, 0.69, 0.69 | 4.41, 4.48, 0.69, 0.68 | 9.95, 9.92, 0.70, 0.70 |
> | std for argmax after smoothing equivariant layers (adopted in the paper)  | 0.03, 0.04, 0.13, 0.11 | 0.36, 0.36, 0.53, 0.56 | 0.33, 0.34, 0.63, 0.65 | 1.13, 1.16, 0.70, 0.69 |
>
> The four numbers in each cell are the estimated std of $x, y, \cos(r), \sin(r)$ correspondingly. The original input values are within $[0, 1]$, and the noise is drawn from $\mathcal{N}(0,\sigma)$ with $\sigma$ specified in the table controlling the noise level. The output $x, y$ are in the range of [0, 63], and $r = 0,\frac{\pi}{2}, \pi \text{ or } \frac{3\pi}{2}$. We use argmax with smoothing equivariant layers in practice, which is relatively stable as shown in the table. The stds of the translation dimension are below one pixel until at least a noise level of $\sigma=0.2$. The rotation dimension is relatively unstable for all versions of $\Phi$, and we will explore preprocessing equivariant layers that could improve its stability in the future. We also notice that the plain argmax function is indeed unstable, which justify the necessity of inserting equivariant layers before taking the argmax.
>
> Furthermore, if we could find a way to differentiate through the sampling index, we could use trainable equivariant layers (possibly parameterised by G-convolutional layers) in $\Phi$. However, how to design such differentiable sampling indices is not trivial, and we consider exploring this in the future.
>
> Overall, we agree with the reviewer that this is an important problem that should be further investigated, there may exist better solutions.
>
> *“The groups explored here, while providing a proof of concept … I would have been much more delighted to see this framework used to tackle e.g. continuous, or at least more fine-grained discrete rotational steps.”*
>
> Indeed it is one of our main future directions to apply this framework empirically to more sophisticated groups such as Lie groups, in particular 3D groups such as SO(3), as mentioned in the Limitations and Future Work section. While the equivariance properties of subsampling/upsampling layers still hold for these cases, we still need to tackle the practical complexities of these applications, e.g. equivariance error that arises due to sampling the signal on G at finitely many group elements (https://arxiv.org/abs/2002.12880, https://arxiv.org/abs/1909.12057). We did not show results with more fine-grained discrete rotational steps because rotations that are not a multiple of 90 degrees would not form a group with discrete translations, although extending our work to continuous translations would address this issue. We leave this as important future work.
>
> *“Given the memorization of shifts, can you envision a way of being able to sample from the GAE latent? Would it require putting a probability measure on the shifts?”*
>
> Note that the equivariant representation $z_{eq}$, which is a tuple of cosets, can be equivalently expressed as a single group element in G (See line 197-200). Therefore, we could specify a prior and posterior over $z_{eq}$ as distributions over $G$. To specify such distributions, one could use a uniform distribution, Mises–Fisher distribution, or more flexible parameterisations such as normalizing flows over tori (https://arxiv.org/pdf/2002.02428.pdf). We think it may be useful to specify an empirical prior rather than a fixed one in this case, because for example, we might want to learn from data the typical orientations of certain objects.

---

> > ### Comment · Reviewer_a1LF · 2021-08-28
> >
> > Thanks for the detailed feedback. I raise my score to 7.
> >
> > Side note: It seems argumentatively slightly inconsistent to exclude finer discrete rotation groups on the grounds of inexact grid sampling, but at the same time perform smoothing to alleviate instability, which also largely alleviates the grid sampling problem.

---

### Decision · Program_Chairs · 2021-09-27

**Decision:**

Accept (Poster)

**Comment:**

This paper introduces a novel method for subsampling along the orbit of a group action, which maintains the equi-variance property. After the rebuttal, all the reviewers agreed that this work is interesting and recommended to accept it. Thus, I will suggest to accept this paper yet the authors should incorporate the relevant suggestions of each reviewer.